# A Trainable Optimal Transport Embedding for Feature Aggregation and its Relationship to Attention

**Grégoire Mialon**[*†‡]**, Dexiong Chen**[*†]**, Alexandre d'Aspremont**[‡] **& Julien Mairal**[†]
[†] Inria, [‡] CNRS-ENS
{gregoire.mialon,dexiong.chen,julien.mairal}@inria.fr, aspremon@ens.fr

## Abstract

We address the problem of learning on sets of features, motivated by the need of performing pooling operations in long biological sequences of varying sizes, with long-range dependencies, and possibly few labeled data. To address this challenging task, we introduce a parametrized representation of fixed size, which embeds and then aggregates elements from a given input set according to the optimal transport plan between the set and a trainable reference. Our approach scales to large datasets and allows end-to-end training of the reference, while also providing a simple unsupervised learning mechanism with small computational cost. Our aggregation technique admits two useful interpretations: it may be seen as a mechanism related to attention layers in neural networks, or it may be seen as a scalable surrogate of a classical optimal transport-based kernel. We experimentally demonstrate the effectiveness of our approach on biological sequences, achieving state-of-the-art results for protein fold recognition and detection of chromatin profiles tasks, and, as a proof of concept, we show promising results for processing natural language sequences. We provide an open-source implementation of our embedding that can be used alone or as a module in larger learning models at https://github.com/claying/OTK.

## 1 Introduction

Many scientific fields such as bioinformatics or natural language processing (NLP) require processing sets of features with positional information (biological sequences, or sentences represented by a set of local features). These objects are delicate to manipulate due to varying lengths and potentially long-range dependencies between their elements. For many tasks, the difficulty is even greater since the sets can be arbitrarily large, or only provided with few labels, or both.

Deep learning architectures specifically designed for sets have recently been proposed (Lee et al., 2019; Skianis et al., 2020). Our experiments show that these architectures perform well for NLP tasks, but achieve mixed performance for long biological sequences of varying size with few labeled data. Some of these models use attention (Bahdanau et al., 2015), a classical mechanism for aggregating features. Its typical implementation is the transformer (Vaswani et al., 2017), which has shown to achieve state-of-the-art results for many sequence modeling tasks, *e.g*, in NLP (Devlin et al., 2019) or in bioinformatics (Rives et al., 2019), when trained with self supervision on large-scale data. Beyond sequence modeling, we are interested in this paper in finding a good representation for sets of features of potentially diverse sizes, with or without positional information, when the amount of training data may be scarce. To this end, we introduce a trainable embedding, which can operate directly on the feature set or be combined with existing deep approaches.

More precisely, our embedding marries ideas from optimal transport (OT) theory (Peyré & Cuturi, 2019) and kernel methods (Schölkopf & Smola, 2001). We call this embedding OTKE (Optimal

---

[*]Equal contribution.

[†]Univ. Grenoble Alpes, Inria, CNRS, Grenoble INP, LJK, 38000 Grenoble, France.

[‡]D.I., UMR 8548, École normale supérieure, Paris, France.

Transport Kernel Embedding). Concretely, we embed feature vectors of a given set to a reproducing kernel Hilbert space (RKHS) and then perform a weighted pooling operation, with weights given by the transport plan between the set and a trainable reference. To gain scalability, we then obtain a finite-dimensional embedding by using kernel approximation techniques (Williams & Seeger, 2001). The motivation for using kernels is to provide a non-linear transformation of the input features before pooling, whereas optimal transport allows to align the features on a trainable reference with fast algorithms (Cuturi, 2013). Such combination provides us with a theoretically grounded, fixed-size embedding that can be learned either without any label, or with supervision. Our embedding can indeed become adaptive to the problem at hand, by optimizing the reference with respect to a given task. It can operate on large sets with varying size, model long-range dependencies when positional information is present, and scales gracefully to large datasets. We demonstrate its effectiveness on biological sequence classification tasks, including protein fold recognition and detection of chromatin profiles where we achieve state-of-the-art results. We also show promising results in natural language processing tasks, where our method outperforms strong baselines.

**Contributions.** In summary, our contribution is three-fold. We propose a new method to embed sets of features of varying sizes to fixed size representations that are well adapted to downstream machine learning tasks, and whose parameters can be learned in either unsupervised or supervised fashion. We demonstrate the scalability and effectiveness of our approach on biological and natural language sequences. We provide an open-source implementation of our embedding that can be used alone or as a module in larger learning models.

## 2    RELATED WORK

**Kernel methods for sets and OT-based kernels.** The kernel associated with our embedding belongs to the family of match kernels (Lyu, 2004; Tolias et al., 2013), which compare all pairs of features between two sets via a similarity function. Another line of research builds kernels by matching features through the Wasserstein distance. A few of them are shown to be positive definite (Gardner et al., 2018) and/or fast to compute (Rabin et al., 2011; Kolouri et al., 2016). Except for few hyper-parameters, these kernels yet cannot be trained end-to-end, as opposed to our embedding that relies on a trainable reference. Efficient and trainable kernel embeddings for biological sequences have also been proposed by Chen et al. (2019a;b). Our work can be seen as an extension of these earlier approaches by using optimal transport rather than mean pooling for aggregating local features, which performs significantly better for long sequences in practice.

**Deep learning for sets.** Deep Sets (Zaheer et al., 2017) feed each element of an input set into a feed-forward neural network. The outputs are aggregated following a simple pooling operation before further processing. Lee et al. (2019) propose a Transformer inspired encoder-decoder architecture for sets which also uses latent variables. Skianis et al. (2020) compute some comparison costs between an input set and reference sets. These costs are then used as features in a subsequent neural network. The reference sets are learned end-to-end. Unlike our approach, such models do not allow unsupervised learning. We will use the last two approaches as baselines in our experiments.

**Interpretations of attention.** Using the transport plan as an ad-hoc attention score was proposed by Chen et al. (2019c) in the context of network embedding to align data modalities. Our paper goes beyond and uses the transport plan as a principle for pooling a set in a model, with trainable parameters. Tsai et al. (2019) provide a view of Transformer's attention via kernel methods, yet in a very different fashion where attention is cast as kernel smoothing and not as a kernel embedding.

## 3    PROPOSED EMBEDDING

### 3.1    PRELIMINARIES

We handle sets of features in $\mathbb{R}^d$ and consider sets $\mathbf{x}$ living in

$$\mathcal{X} = \left\{ \mathbf{x} \mid \mathbf{x} = \{\mathbf{x}_1, \dots, \mathbf{x}_n\} \text{ such that } \mathbf{x}_1, \dots, \mathbf{x}_n \in \mathbb{R}^d \text{ for some } n \geq 1 \right\}.$$

Elements of $\mathcal{X}$ are typically vector representations of local data structures, such as $k$-mers for sequences, patches for natural images, or words for sentences. The size of $\mathbf{x}$ denoted by $n$ may vary, which is not an issue since the methods we introduce may take a sequence of any size as input, while providing a fixed-size embedding. We now revisit important results on optimal transport and kernel methods, which will be useful to describe our embedding and its computation algorithms.

**Optimal transport.** Our pooling mechanism will be based on the transport plan between $\mathbf{x}$ and $\mathbf{x}'$ seen as weighted point clouds or discrete measures, which is a by-product of the optimal transport problem (Villani, 2008; Peyré & Cuturi, 2019). OT has indeed been widely used in alignment problems (Grave et al., 2019). Throughout the paper, we will refer to the Kantorovich relaxation of OT with entropic regularization, detailed for example in (Peyré & Cuturi, 2019). Let $\mathbf{a}$ in $\Delta^n$ (probability simplex) and $\mathbf{b}$ in $\Delta^{n'}$ be the weights of the discrete measures $\sum_i \mathbf{a}_i \delta_{\mathbf{x}_i}$ and $\sum_j \mathbf{b}_j \delta_{\mathbf{x}'_j}$ with respective locations $\mathbf{x}$ and $\mathbf{x}'$, where $\delta_{\mathbf{x}}$ is the Dirac at position $\mathbf{x}$. Let $\mathbf{C}$ in $\mathbb{R}^{n \times n'}$ be a matrix representing the pairwise costs for aligning the elements of $\mathbf{x}$ and $\mathbf{x}'$. The entropic regularized Kantorovich relaxation of OT from $\mathbf{x}$ to $\mathbf{x}'$ is

$$\min_{\mathbf{P} \in U(\mathbf{a},\mathbf{b})} \sum_{ij} \mathbf{C}_{ij} \mathbf{P}_{ij} - \varepsilon H(\mathbf{P}), \tag{1}$$

where $H(\mathbf{P}) = -\sum_{ij} \mathbf{P}_{ij}(\log(\mathbf{P}_{ij}) - 1)$ is the entropic regularization with parameter $\varepsilon$, which controls the sparsity of $\mathbf{P}$, and $U$ is the space of admissible couplings between $\mathbf{a}$ and $\mathbf{b}$:

$$U(\mathbf{a}, \mathbf{b}) = \{\mathbf{P} \in \mathbb{R}_+^{n \times n'} : \mathbf{P}\mathbf{1}_n = \mathbf{a} \text{ and } \mathbf{P}^\top \mathbf{1}_{n'} = \mathbf{b}\}.$$

The problem is typically solved by using a matrix scaling procedure known as Sinkhorn's algorithm (Sinkhorn & Knopp, 1967; Cuturi, 2013). In practice, $\mathbf{a}$ and $\mathbf{b}$ are uniform measures since we consider the mass to be evenly distributed between the points. $\mathbf{P}$ is called the transport plan, which carries the information on how to distribute the mass of $\mathbf{x}$ in $\mathbf{x}'$ with minimal cost. Our method uses optimal transport to align features of a given set to a learned reference.

**Kernel methods.** Kernel methods (Schölkopf & Smola, 2001) map data living in a space $\mathcal{X}$ to a reproducing kernel Hilbert space $\mathcal{H}$, associated to a positive definite kernel $K$ through a mapping function $\varphi : \mathcal{X} \to \mathcal{H}$, such that $K(\mathbf{x}, \mathbf{x}') = \langle \varphi(\mathbf{x}), \varphi(\mathbf{x}') \rangle_{\mathcal{H}}$. Even though $\varphi(\mathbf{x})$ may be infinite-dimensional, classical kernel approximation techniques (Williams & Seeger, 2001) provide finite-dimensional embeddings $\psi(\mathbf{x})$ in $\mathbb{R}^k$ such that $K(\mathbf{x}, \mathbf{x}') \approx \langle \psi(\mathbf{x}), \psi(\mathbf{x}') \rangle$. Our embedding for sets relies in part on kernel method principles and on such a finite-dimensional approximation.

## 3.2 Optimal Transport Embedding and Associated Kernel

We now present the OTKE, an embedding and pooling layer which aggregates a variable-size set or sequence of features into a fixed-size embedding. We start with an infinite-dimensional variant living in a RKHS, before introducing the finite-dimensional embedding that we use in practice.

**Infinite-dimensional embedding in RKHS.** Given a set $\mathbf{x}$ and a (learned) reference $\mathbf{z}$ in $\mathcal{X}$ with $p$ elements, we consider an embedding $\Phi_{\mathbf{z}}(\mathbf{x})$ which performs the following operations: (i) initial embedding of the elements of $\mathbf{x}$ and $\mathbf{z}$ to a RKHS $\mathcal{H}$; (ii) alignment of the elements of $\mathbf{x}$ to the elements of $\mathbf{z}$ via optimal transport; (iii) weighted linear pooling of the elements $\mathbf{x}$ into $p$ bins, producing an embedding $\Phi_{\mathbf{z}}(\mathbf{x})$ in $\mathcal{H}^p$, which is illustrated in Figure 1.

Before introducing more formal details, we note that our embedding relies on two main ideas:

- *Global similarity-based pooling using references.* Learning on large sets with long-range interactions may benefit from pooling to reduce the number of feature vectors. Our pooling rule follows an inductive bias akin to that of self-attention: elements that are relevant to each other for the task at hand should be pooled together. To this end, each element in the reference set corresponds to a pooling cell, where the elements of the input set are aggregated through a weighted sum. The weights simply reflect the similarity between the vectors of the input set and the current vector in the reference. Importantly, using a reference set enables to reduce the size of the "attention matrix" from quadratic to linear in the length of the input sequence.
- *Computing similarity weights via optimal transport.* A computationally efficient similarity score between two elements is their dot-product (Vaswani et al., 2017). In this paper, we rather consider that elements of the input set should be pooled together if they align well with the same part of the reference. Alignment scores can efficiently be obtained by computing the transport plan between the input and the reference sets: Sinkhorn's algorithm indeed enjoys fast solvers (Cuturi, 2013).

We are now in shape to give a formal definition.

**Definition 3.1 (The optimal transport kernel embedding).** Let $\mathbf{x} = (\mathbf{x}_1, \ldots, \mathbf{x}_n)$ in $\mathcal{X}$ be an input set of feature vectors and $\mathbf{z} = (\mathbf{z}_1, \ldots, \mathbf{z}_p)$ in $\mathcal{X}$ be a reference set with $p$ elements. Let $\kappa$ be a

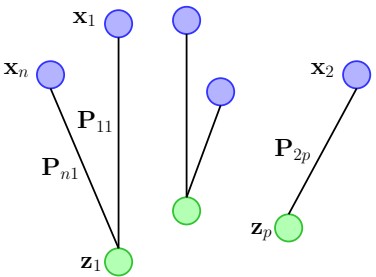 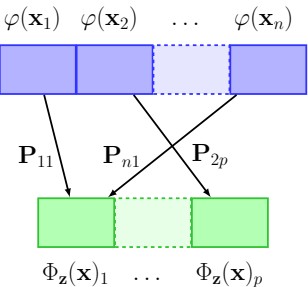

Figure 1: The input point cloud $\mathbf{x}$ is transported onto the reference $\mathbf{z} = (\mathbf{z}_1, \ldots, \mathbf{z}_p)$ (left), yielding the optimal transport plan $\mathbf{P}_\kappa(\mathbf{x}, \mathbf{z})$ used to aggregate the embedded features and form $\Phi_\mathbf{z}(\mathbf{x})$ (right).

positive definite kernel, *e.g.*, Gaussian kernel, with RKHS $\mathcal{H}$ and $\varphi : \mathbb{R}^d \to \mathcal{H}$, its associated kernel embedding. Let $\boldsymbol{\kappa}$ be the $n \times p$ matrix which carries the comparisons $\kappa(\mathbf{x}_i, \mathbf{z}_j)$, before alignment.

Then, the transport plan between $\mathbf{x}$ and $\mathbf{z}$, denoted by the $n \times p$ matrix $\mathbf{P}(\mathbf{x}, \mathbf{z})$, is defined as the unique solution of (1) when choosing the cost $\mathbf{C} = -\boldsymbol{\kappa}$, and our embedding is defined as

$$\Phi_\mathbf{z}(\mathbf{x}) := \sqrt{p} \times \left( \sum_{i=1}^n \mathbf{P}(\mathbf{x}, \mathbf{z})_{i1} \varphi(\mathbf{x}_i), \ \ldots, \ \sum_{i=1}^n \mathbf{P}(\mathbf{x}, \mathbf{z})_{ip} \varphi(\mathbf{x}_i) \right) = \sqrt{p} \times \mathbf{P}(\mathbf{x}, \mathbf{z})^\top \varphi(\mathbf{x}),$$

where $\varphi(\mathbf{x}) := [\varphi(\mathbf{x}_1), \ldots, \varphi(\mathbf{x}_n)]^\top$.

Interestingly, it is easy to show that our embedding $\Phi_\mathbf{z}(\mathbf{x})$ is associated to the positive definite kernel

$$K_\mathbf{z}(\mathbf{x}, \mathbf{x}') := \sum_{i,i'=1}^n \mathbf{P}_\mathbf{z}(\mathbf{x}, \mathbf{x}')_{ii'} \kappa(\mathbf{x}_i, \mathbf{x}'_{i'}) = \langle \Phi_\mathbf{z}(\mathbf{x}), \Phi_\mathbf{z}(\mathbf{x}') \rangle, \tag{2}$$

with $\mathbf{P}_\mathbf{z}(\mathbf{x}, \mathbf{x}') := p \times \mathbf{P}(\mathbf{x}, \mathbf{z}) \mathbf{P}(\mathbf{x}', \mathbf{z})^\top$. This is a weighted match kernel, with weights given by optimal transport in $\mathcal{H}$. The notion of pooling in the RKHS $\mathcal{H}$ of $\kappa$ arises naturally if $p \leq n$. The elements of $\mathbf{x}$ are non-linearly embedded and then aggregated in "buckets", one for each element in the reference $\mathbf{z}$, given the values of $\mathbf{P}(\mathbf{x}, \mathbf{z})$. This process is illustrated in Figure 1. We acknowledge here the concurrent work by Kolouri et al. (2021), where a similar embedding is used for graph representation. We now expose the benefits of this kernel formulation, and its relation to classical non-positive definite kernel.

**Kernel interpretation.** Thanks to the gluing lemma (see, *e.g.*, Peyré & Cuturi, 2019), $\mathbf{P}_\mathbf{z}(\mathbf{x}, \mathbf{x}')$ is a valid transport plan and, empirically, a rough approximation of $\mathbf{P}(\mathbf{x}, \mathbf{x}')$. $K_\mathbf{z}$ can therefore be seen as a surrogate of a well-known kernel (Rubner et al., 2000), defined as

$$K_{\mathrm{OT}}(\mathbf{x}, \mathbf{x}') := \sum_{i,i'=1}^n \mathbf{P}(\mathbf{x}, \mathbf{x}')_{ii'} \kappa(\mathbf{x}_i, \mathbf{x}'_{i'}). \tag{3}$$

When the entropic regularization $\varepsilon$ is equal to 0, $K_{\mathrm{OT}}$ is equivalent to the 2-Wasserstein distance $W_2(\mathbf{x}, \mathbf{x}')$ with ground metric $d_\kappa$ induced by kernel $\kappa$. $K_{\mathrm{OT}}$ is generally not positive definite (see Peyré & Cuturi (2019), Chapter 8.3) and computationally costly (the number of transport plans to compute is quadratic in the number of sets to process whereas it is linear for $K_\mathbf{z}$). Now, we show the relationship between this kernel and our kernel $K_\mathbf{z}$, which is proved in Appendix B.1.

**Lemma 3.1** (Relation between $\mathbf{P}(\mathbf{x}, \mathbf{x}')$ and $\mathbf{P}_\mathbf{z}(\mathbf{x}, \mathbf{x}')$ when $\varepsilon = 0$). For any $\mathbf{x}, \mathbf{x}'$ and $\mathbf{z}$ in $\mathcal{X}$ with lengths $n$, $n'$ and $p$, by denoting $W_2^\mathbf{z}(\mathbf{x}, \mathbf{x}') := \langle \mathbf{P}_\mathbf{z}(\mathbf{x}, \mathbf{x}'), d_\kappa^2(\mathbf{x}, \mathbf{x}') \rangle^{1/2}$ we have

$$|W_2(\mathbf{x}, \mathbf{x}') - W_2^\mathbf{z}(\mathbf{x}, \mathbf{x}')| \leq 2 \min(W_2(\mathbf{x}, \mathbf{z}), W_2(\mathbf{x}', \mathbf{z})). \tag{4}$$

This lemma shows that the distance $W_2^\mathbf{z}$ resulting from $K_\mathbf{z}$ is related to the Wasserstein distance $W_2$; yet, this relation should not be interpreted as an approximation error as our goal is not to approximate $W_2$, but rather to derive a trainable embedding $\Phi_\mathbf{z}(\mathbf{x})$ with good computational properties. Lemma 3.1 roots our features and to some extent self-attention in a rich optimal transport literature. In fact, $W_2^\mathbf{z}$ is equivalent to a distance introduced by Wang et al. (2013), whose properties are further studied by Moosmüller & Cloninger (2020). A major difference is that $W_2^\mathbf{z}$ crucially relies on Sinkhorn's algorithm so that the references can be learned end-to-end, as explained below.

### 3.3 FROM INFINITE-DIMENSIONAL KERNEL EMBEDDING TO FINITE DIMENSION

In some cases, $\varphi(\mathbf{x})$ is already finite-dimensional, which allows to compute the embedding $\Phi_{\mathbf{z}}(\mathbf{x})$ explicitly. This is particularly useful when dealing with large-scale data, as it enables us to use our method for supervised learning tasks without computing the Gram matrix, which grows quadratically in size with the number of samples. When $\varphi$ is infinite or high-dimensional, it is nevertheless possible to use an approximation based on the Nyström method (Williams & Seeger, 2001), which provides an embedding $\psi : \mathbb{R}^d \to \mathbb{R}^k$ such that

$$\langle \psi(\mathbf{x}_i), \psi(\mathbf{x}'_j) \rangle_{\mathbb{R}^k} \approx \kappa(\mathbf{x}_i, \mathbf{x}'_j).$$

Concretely, the Nyström method consists in projecting points from the RKHS $\mathcal{H}$ onto a linear subspace $\mathcal{F}$, which is parametrized by $k$ anchor points $\mathcal{F} = \text{Span}(\varphi(\mathbf{w}_1), \ldots, \varphi(\mathbf{w}_k))$. The corresponding embedding admits an explicit form $\psi(\mathbf{x}_i) = \kappa(\mathbf{w}, \mathbf{w})^{-1/2} \kappa(\mathbf{w}, \mathbf{x}_i)$, where $\kappa(\mathbf{w}, \mathbf{w})$ is the $k \times k$ Gram matrix of $\kappa$ computed on the set $\mathbf{w} = \{\mathbf{w}_1, \ldots, \mathbf{w}_k\}$ of anchor points and $\kappa(\mathbf{w}, \mathbf{x}_i)$ is in $\mathbb{R}^k$. Then, there are several ways to learn the anchor points: (a) they can be chosen as random points from data; (b) they can be defined as centroids obtained by K-means, see Zhang et al. (2008); (c) they can be learned by back-propagation for a supervised task, see Mairal (2016).

Using such an approximation within our framework can be simply achieved by (i) replacing $\kappa$ by a linear kernel and (ii) replacing each element $\mathbf{x}_i$ by its embedding $\psi(\mathbf{x}_i)$ in $\mathbb{R}^k$ and considering a reference set with elements in $\mathbb{R}^k$. By abuse of notation, we still use $\mathbf{z}$ for the new parametrization. The embedding, which we use in practice in all our experiments, becomes simply

$$\Phi_{\mathbf{z}}(\mathbf{x}) = \sqrt{p} \times \left( \sum_{i=1}^{n} \mathbf{P}(\psi(\mathbf{x}), \mathbf{z})_{i1} \psi(\mathbf{x}_i), \ \ldots, \ \sum_{i=1}^{n} \mathbf{P}(\psi(\mathbf{x}), \mathbf{z})_{ip} \psi(\mathbf{x}_i) \right)$$
$$= \sqrt{p} \times \mathbf{P}(\psi(\mathbf{x}), \mathbf{z})^{\top} \psi(\mathbf{x}) \in \mathbb{R}^{p \times k}, \tag{5}$$

where $p$ is the number of elements in $\mathbf{z}$. Next, we discuss how to learn the reference set $\mathbf{z}$.

### 3.4 UNSUPERVISED AND SUPERVISED LEARNING OF PARAMETER $\mathbf{z}$

**Unsupervised learning.** In the fashion of the Nyström approximation, the $p$ elements of $\mathbf{z}$ can be defined as the centroids obtained by K-means applied to all features from training sets in $\mathcal{X}$. A corollary of Lemma 3.1 suggests another algorithm: a bound on the deviation term between $W_2$ and $W_2^{\mathbf{z}}$ for $m$ samples $(\mathbf{x}^1, \ldots, \mathbf{x}^m)$ is indeed

$$\mathcal{E}^2 := \frac{1}{m^2} \sum_{i,j=1}^{m} |W_2(\mathbf{x}^i, \mathbf{x}^j) - W_2^{\mathbf{z}}(\mathbf{x}^i, \mathbf{x}^j)|^2 \leq \frac{4}{m} \sum_{i=1}^{m} W_2^2(\mathbf{x}^i, \mathbf{z}). \tag{6}$$

The right-hand term corresponds to the objective of the Wasserstein barycenter problem (Cuturi & Doucet, 2013), which yields the mean of a set of empirical measures (here the $\mathbf{x}$'s) under the OT metric. The Wasserstein barycenter is therefore an attractive candidate for choosing $\mathbf{z}$. K-means can be seen as a particular case of Wasserstein barycenter when $m = 1$ (Cuturi & Doucet, 2013; Ho et al., 2017) and is faster to compute. In practice, both methods yield similar results, see Appendix C, and we thus chose K-means to learn $\mathbf{z}$ in unsupervised settings throughout the experiments. The anchor points $\mathbf{w}$ and the references $\mathbf{z}$ may be then computed using similar algorithms; however, their mathematical interpretation differs as exposed above. The task of representing features (learning $\mathbf{w}$ in $\mathbb{R}^d$ for a specific $\kappa$) is decoupled from the task of aggregating (learning the reference $\mathbf{z}$ in $\mathbb{R}^k$).

**Supervised learning.** As mentioned in Section 3.1, $\mathbf{P}(\psi(\mathbf{x}), \mathbf{z})$ is computed using Sinkhorn's algorithm, recalled in Appendix A, which can be easily adapted to batches of samples $\mathbf{x}$, with possibly varying lengths, leading to GPU-friendly forward computations of the embedding $\Phi_{\mathbf{z}}$. More important, all Sinkhorn's operations are differentiable, which enables $\mathbf{z}$ to be optimized with stochastic gradient descent through back-propagation (Genevay et al., 2018), *e.g.*, for minimizing a classification or regression loss function when labels are available. In practice, a small number of Sinkhorn iterations (*e.g.*, 10) is sufficient to compute $\mathbf{P}(\psi(\mathbf{x}), \mathbf{z})$. Since the anchors $\mathbf{w}$ in the embedding layer below can also be learned end-to-end (Mairal, 2016), our embedding can thus be used as a module injected into any model, *e.g*, a deep network, as demonstrated in our experiments.

## 3.5 EXTENSIONS

**Integrating positional information into the embedding.** The discussed embedding and kernel do not take the position of the features into account, which may be problematic when dealing with structured data such as images or sentences. To this end, we borrow the idea of convolutional kernel networks, or CKN (Mairal, 2016; Mairal et al., 2014), and penalize the similarities exponentially with the positional distance between a pair of elements in the input and reference sequences. More precisely, we multiply $\mathbf{P}(\psi(\mathbf{x}), \mathbf{z})$ element-wise by a distance matrix $\mathbf{S}$ defined as:

$$\mathbf{S}_{ij} = e^{-\frac{1}{\sigma_{\text{pos}}^2}(i/n - j/p)^2},$$

and replace it in the embedding. With similarity weights based *both* on content and position, the kernel associated to our embedding can be viewed as a generalization of the CKNs (whose similarity weights are based on position only), with feature alignment based on optimal transport. When dealing with multi-dimensional objects such as images, we just replace the index scalar $i$ with an index vector of the same spatial dimension as the object, representing the positions of each dimension.

**Using multiple references.** A naive reconstruction using different references $\mathbf{z}^1, \ldots, \mathbf{z}^q$ in $\mathcal{X}$ may yield a better approximation of the transport plan. In this case, the embedding of $\mathbf{x}$ becomes

$$\Phi_{\mathbf{z}^1, \ldots, \mathbf{z}^q}(\mathbf{x}) = 1/\sqrt{q} \left( \Phi_{\mathbf{z}^1}(\mathbf{x}), \ldots, \Phi_{\mathbf{z}^q}(\mathbf{x}) \right), \tag{7}$$

with $q$ the number of references (the factor $1/\sqrt{q}$ comes from the mean). Using Eq. (4), we can obtain a bound similar to (6) for a data set of $m$ samples $(\mathbf{x}^1, \ldots, \mathbf{x}^m)$ and $q$ references (see Appendix B.2 for details). To choose multiple references, we tried a K-means algorithm with 2-Wasserstein distance for assigning clusters, and we updated the centroids as in the single-reference case. Using multiple references appears to be useful when optimizing $\mathbf{z}$ with supervision (see Appendix C).

## 4 RELATION BETWEEN OUR EMBEDDING AND SELF-ATTENTION

Our embedding and a single layer of transformer encoder, recalled in Appendix A, share the same type of inductive bias, *i.e*, aggregating features relying on similarity weights. We now clarify their relationship. Our embedding is arguably simpler (see respectively size of attention and number of parameters in Table 1), and may compete in some settings with the transformer self-attention as illustrated in Section 5.

**Shared reference versus self-attention.** There is a correspondence between the values, attention matrix in the transformer and $\varphi$, $\mathbf{P}$ in Definition 3.1, yet also noticeable differences. On the one hand, $\Phi_{\mathbf{z}}$ aligns a given sequence $\mathbf{x}$ with respect to a reference $\mathbf{z}$, learned with or without supervision, and shared across the data set. Our weights are computed using optimal transport. On the other hand, a transformer encoder performs self-alignment: for a given $\mathbf{x}_i$, features are aggregated depending on a similarity score between $\mathbf{x}_i$ and the elements of $\mathbf{x}$ only. The similarity score is a matrix product between queries $Q$ and keys $K$ matrices, learned with supervision and shared across the data set. In this regard, our work complements a recent line of research questioning the dot-product, learned self-attention (Raganato et al., 2020; Weiqiu et al., 2020). Self-attention-like weights can also be obtained with our embedding by computing $\mathbf{P}(\mathbf{x}, \mathbf{z}_i)\mathbf{P}(\mathbf{x}, \mathbf{z}_i)^\top$ for each reference $i$. In that sense, our work is related to recent research on efficient self-attention (Wang et al., 2020; Choromanski et al., 2020), where a low-rank approximation of the self-attention matrix is computed.

Table 1: Relationship between $\Phi_{\mathbf{z}}$ and transformer self-attention. $k$: a function describing how the transformer integrates positional information; $n$: sequence length; $q$: number of references or attention heads; $d$: dimension of the embeddings; $p$: number of supports in $\mathbf{z}$. Typically, $p \ll d$. In recent transformer architectures, positional encoding requires learning additional parameters ($\sim qd^2$).

|  | Self-Attention | $\Phi_{\mathbf{z}}$ |
|---|---|---|
| Attention score | $\mathbf{W} = W^\top Q$ | $\mathbf{P}$ |
| Size of score | $O(n^2)$ | $O(np)$ |
| Alignment w.r.t: | $\mathbf{x}$ itself | $\mathbf{z}$ |
| Learned + Shared | $W$ and $Q$ | $\mathbf{z}$ |
| Nonlinear mapping | Feed-forward | $\varphi$ or $\psi$ |
| Position encoding | $k(t_i, t'_j)$ | $e^{-\frac{1}{\sigma_{\text{pos}}^2}(\frac{i}{n} - \frac{j}{n'})^2}$ |
| Nb. parameters | $\sim qd^2$ | $qpd$ |
| Supervision | Needed | Not needed |

**Position smoothing and relative positional encoding.** Transformers can add an absolute positional encoding to the input features (Vaswani et al., 2017). Yet, relative positional encoding (Dai et al., 2019) is a current standard for integrating positional information: the position offset between the query element and a given key can be injected in the attention score (Tsai et al., 2019), which is equivalent to our approach. The link between CKNs and our kernel, provided by this positional encoding, stands in line with recent works casting attention and convolution into a unified framework (Andreoli, 2019). In particular, Cordonnier et al. (2020) show that attention learns convolution in the setting of image classification: the kernel pattern is learned at the same time as the filters.

**Multiple references and attention heads.** In the transformer architecture, the succession of blocks composed of an attention layer followed by a fully-connected layer is called a head, with each head potentially focusing on different parts of the input. Successful architectures have a few heads in parallel. The outputs of the heads are then aggregated to output a final embedding. A layer of our embedding with non-linear kernel $\kappa$ can be seen as such a block, with the references playing the role of the heads. As some recent works question the role of attention heads (Voita et al., 2019; Michel et al., 2019), exploring the content of our learned references $\mathbf{z}$ may provide another perspective on this question. More generally, visualization and interpretation of the learned references could be of interest for biological sequences.

## 5 EXPERIMENTS

We now show the effectiveness of our embedding OTKE in tasks where samples can be expressed as large sets with potentially few labels, such as in bioinformatics. We evaluate our embedding alone in unsupervised or supervised settings, or within a model in the supervised setting. We also consider NLP tasks involving shorter sequences and relatively more labels.

### 5.1 DATASETS, EXPERIMENTAL SETUP AND BASELINES

In unsupervised settings, we train a linear classifier with the cross entropy loss between true labels and predictions on top of the features provided by our embedding (where the references $\mathbf{z}$ and Nyström anchors $\mathbf{w}$ have been learned without supervision), or an unsupervised baseline. In supervised settings, the same model is initialized with our unsupervised method and then trained end-to-end (including $\mathbf{z}$ and $\mathbf{w}$) by minimizing the same loss. We use an alternating optimization strategy to update the parameters for both SCOP and SST datasets, as used by Chen et al. (2019a;b). We train for 100 epochs with Adam on both data sets: the initial learning rate is 0.01, and get halved as long as there is no decrease in the validation loss for 5 epochs. The hyper-parameters we tuned include number of supports and references $p, q$, entropic regularization in OT $\varepsilon$, the bandwidth of Gaussian kernels and the regularization parameter of the linear classifier. The best values of $\varepsilon$ and the bandwidth were found stable across tasks, while the regularization parameter needed to be more carefully cross-validated. Additional results and implementation details can be found in Appendix C.

**Protein fold classification on SCOP 1.75.** We follow the protocol described by Hou et al. (2019) for this important task in bioinformatics. The dataset contains $19,245$ sequences from $1,195$ different classes of fold (hence less than 20 labels in average per class). The sequence lengths vary from tens to thousands. Each element of a sequence is a $45$-dimensional vector. The objective is to classify the sequences to fold classes, which corresponds to a multiclass classification problem. The features fed to the linear classifier are the output of our embedding with $\varphi$ the Gaussian kernel mapping on $k$-mers (subsequences of length $k$) with $k$ fixed to be 10, which is known to perform well in this task (Chen et al., 2019a). The number of anchor points for Nyström method is fixed to 1024 and 512 respectively for unsupervised and supervised setting. In the unsupervised setting, we compare our method to state-of-the-art unsupervised method for this task: CKN (Chen et al., 2019a), which performs a global mean pooling in contrast to the global adaptive pooling performed by our embedding. In the supervised setting, we compare the same model to the following supervised models: CKN, Recurrent Kernel Networks (RKN) (Chen et al., 2019b), a CNN with 10 convolutional layers named DeepSF (Hou et al., 2019), Rep the Set (Skianis et al., 2020) and Set Transformer (Lee et al., 2019), using the public implementations by their authors. Rep the Set and Set Transformer are used on the top of a convolutional layer of the same filter size as CKN to extract $k$-mer features. Their model hyper-parameters, weight decay and learning rate are tuned in the same way as for our models

(see Appendix for details). The default architecture of Set Transformer did not perform well due to overfitting. We thus used a shallower architecture with one Induced Set Attention Block (ISAB), one Pooling by Multihead Attention (PMA) and one linear layer, similar to the one-layer architectures of CKN and our model. The results are shown in Table 2.

Table 2: Classification accuracy (top 1/5/10) on test set for SCOP 1.75 for different unsupervised and supervised baselines, averaged from 10 different runs ($q$ references $\times$ $p$ supports).

| Method | Unsupervised | Supervised |
|---|---|---|
| DeepSF (Hou et al., 2019) | Not available. | 73.0/90.3/94.5 |
| CKN (Chen et al., 2019a) | 81.8±0.8/92.8±0.2/95.0±0.2 | 84.1±0.1/94.3±0.2/96.4±0.1 |
| RKN (Chen et al., 2019b) | Not available. | 85.3±0.3/95.0±0.2/96.5±0.1 |
| Set Transformer (Lee et al., 2019) | Not available. | 79.2±4.6/91.5±1.4/94.3±0.6 |
| Approximate Rep the Set (Skianis et al., 2020) | Not available. | 84.5±0.6/94.0±0.4/95.7±0.4 |
| Ours (dot-product instead of OT) | 78.2±1.9/93.1±0.7/96.0±0.4 | 87.5±0.3/95.5±0.2/96.9±0.1 |
| Ours (Unsup.: $1 \times 100$ / Sup.: $5 \times 10$) | **85.8±0.2/95.3±0.1/96.8±0.1** | **88.7±0.3/95.9±0.2/97.3±0.1** |

**Detection of chromatin profiles.** Predicting the chromatin features such as transcription factor (TF) binding from raw genomic sequences has been studied extensively in recent years. CNNs with max pooling operations have been shown effective for this task. Here, we consider DeepSEA dataset (Zhou & Troyanskaya, 2015) consisting in simultaneously predicting 919 chromatin profiles, which can be formulated as a multi-label classification task. DeepSEA contains $4,935,024$ DNA sequences of length 1000 and each of them is associated with 919 different labels (chromatin profiles). Each sequence is represented as a $1000 \times 4$ binary matrix through one-hot encoding and the objective is to predict which profiles a given sequence possesses. As this problem is very imbalanced for each profile, learning an unsupervised model could require an extremely large number of parameters. We thus only consider our supervised embedding as an adaptive pooling layer and inject it into a deep neural network, between one convolutional layer and one fully connected layer, as detailed in Appendix C.4. In our embedding, $\varphi$ is chosen to be identity and the positional encoding described in Section 3 is used. We compare our model to a state-of-the-art CNN with 3 convolutional layers and two fully-connected layers (Zhou & Troyanskaya, 2015). The results are shown in Table 3.

**Sentiment analysis on Stanford Sentiment Treebank.** SST-2 (Socher et al., 2013) belongs to the NLP GLUE benchmark (Wang et al., 2019) and consists in predicting whether a movie review is positive. The dataset contains $70,042$ reviews. The test predictions need to be submitted on the GLUE leaderboard, so that we remove a portion of the training set for validation purpose and report accuracies on the actual validation set used as a test set. Our model is one layer of our embedding with $\varphi$ a Gaussian kernel mapping with 64 Nyström filters in the supervised setting, and a linear mapping in the unsupervised setting. The features used in our model and all baselines are word vectors with dimension 768 provided by the HuggingFace implementation (Wolf et al., 2019) of the transformer BERT (Devlin et al., 2019). State-of-the-art accuracies are usually obtained after supervised fine-tuning of pre-trained transformers. Training a linear model on pre-trained features after simple pooling (*e.g*, mean) also yields good results. [CLS], which denotes the BERT embedding used for classification, is also a common baseline. The results are shown in Table 4.

## 5.2 RESULTS AND DISCUSSION

In protein fold classification, our embedding outperforms all baselines in both unsupervised and supervised settings. Surprisingly, our unsupervised model also achieves better results than most supervised baselines. In contrast, Set Transformer does not perform well, possibly because its implementation was not designed for sets with varying sizes, and tasks with few annotations. In detection of chromatin profiles, our model (our embedding within a deep network) has fewer layers than state-of-the-art CNNs while outperforming them, which advocates for the use of attention-based models for such applications. Our results also suggest that positional information is important

Table 3: Results for prediction of chromatin profiles on the DeepSEA dataset. The metrics are area under ROC (auROC) and area under PR curve (auPRC), averaged over 919 chromatin profiles. Due to the huge size of the dataset, we only provide results based on a single run.

| Method | auROC | auPRC |
|---|---|---|
| DeepSEA (Zhou & Troyanskaya, 2015) | 0.933 | 0.342 |
| Ours with position encoding (Sinusoidal (Vaswani et al., 2017)/Ours) | 0.917/**0.936** | 0.311/**0.360** |

Table 4: Classification accuracies for SST-2 reported on standard validation set, averaged from 10 different runs ($q$ references $\times$ $p$ supports).

| Method | Unsupervised | Supervised |
|---|---|---|
| [CLS] embedding (Devlin et al., 2019) | 84.6±0.3 | 90.3±0.1 |
| Mean Pooling of BERT features (Devlin et al., 2019) | 85.3±0.4 | **90.8±0.1** |
| Approximate Rep the Set (Skianis et al., 2020) | Not available. | 86.8±0.9 |
| Rep the Set (Skianis et al., 2020) | Not available. | 87.1±0.5 |
| Set Transformer (Lee et al., 2019) | Not available. | 87.9±0.8 |
| Ours (dot-product instead of OT) | 85.7±0.9 | 86.9±1.1 |
| Ours (Unsupervised: $1 \times 300$. Supervised: $4 \times 30$) | **86.8±0.3** | 88.1±0.8 |

(Appendix C.4;C.2), and our Gaussian position encoding outperforms the sinusoidal one introduced in Vaswani et al. (2017). Note that in contrast to a typical transformer, which would have stored a $1000 \times 1000$ attention matrix, our attention score with a reference of size $64$ is only $1000 \times 64$, which illustrates the discussion in Section 4. In NLP, an *a priori* less favorable setting since sequences are shorter and there are more data, our supervised embedding gets close to a strong state-of-the-art, *i.e.* a fully-trained transformer. We observed our method to be much faster than RepSet, as fast as Set Transformer, yet slower than ApproxRepSet (C.3). Using the OT plan as similarity score yields better accuracies than the dot-product between the input sets and the references (see Table 2; 4).

**Choice of parameters.** This paragraph sums up the impact of hyper-parameter choices. Experiments justifying our claims can be found in Appendix C.

- Number of references $q$: for biological sequences, a single reference was found to be enough in the unsupervised case, see Table 11. In the supervised setting, Table 14 suggests that using $q = 5$ provides slightly better results but $q = 1$ remains a good baseline, and that the sensitivity to number of references is moderate.
- Number of supports $p$ in a reference: Table 11 and Table 14 suggest that the sensitivity of the model to the number of supports is also moderate.
- Nyström anchors: an anchor can be seen as a neuron in a feed-forward neural network (see expression of $\psi$ in 3.3). In unsupervised settings, the more anchors, the better the approximation of the kernel matrix. Then, the performance saturates, see Table 12. In supervised settings, the optimal number of anchors points is much smaller, as also observed by Chen et al. (2019a), Fig 6.
- Bandwidth $\sigma$ in gaussian kernel: $\sigma$ was chosen as in Chen et al. (2019b) and we did not try to optimize it in this work, as it seemed to already provide good results. Nevertheless, slightly better results can be obtained when tuning this parameter, for instance in SST-2.

**OTKE and self-supervised methods.** Our approach should not be positioned against self-supervision and instead brings complementary features: the OTKE may be plugged in state-of-the-art models pre-trained on large unannotated corpus. For instance, on SCOP 1.75, we use ESM-1 (Rives et al., 2019), pretrained on 250 millions protein sequences, with mean pooling followed by a linear classifier. As we do not have the computational ressources to fine-tune ESM1-t34, we only train a linear layer on top of the extracted features. Using the same model, we replace the mean pooling by our (unsupervised) OTKE layer, and also only train the linear layer. This results in accuracy improvements as showed in Table 5. While training huge self-supervised learning models on large datasets is very effective, ESM1-t34 admits more than 2500 times more parameters than our single-layer OTKE model (260k parameters versus 670M) and our single-layer OTKE outperforms smaller versions of ESM1 (43M parameters). Finally, self-supervised pre-training of a deep model including OTKE on large data sets would be interesting for fair comparison.

Table 5: Classification accuracy (top 1/5/10) results of our unsupervised embedding for SCOP 1.75 with pre-trained ESM models (Rives et al., 2019).

| Model | Nb parameters | Mean Pooling | Unsupervised OTKE |
|---|---|---|---|
| ESM1-t6-43M-UR50S | 43M | 84.01/93.17/95.07 | 85.91/93.72/95.30 |
| ESM1-t34-670M-UR50S | 670M | 94.95/97.32/97.91 | 95.22/97.32/98.03 |

**Multi-layer extension.** Extending the OTKE to a multi-layer embedding is a natural yet not straightforward research direction: it is not clear how to find a right definition of intermediate feature aggregation in a multi-layer OTKE model. Note that for DeepSEA, our model with single-layer OTKE already outperforms a multi-layer CNN, which suggests that a multi-layer OTKE is not always needed.

ACKNOWLEDGMENTS

JM, GM and DC were supported by the ERC grant number 714381 (SOLARIS project) and by ANR 3IA MIAI@Grenoble Alpes, (ANR-19-P3IA-0003). AA would like to acknowledge support from the *ML and Optimisation* joint research initiative with the *fonds AXA pour la recherche* and Kamet Ventures, a Google focused award, as well as funding by the French government under management of Agence Nationale de la Recherche as part of the "Investissements d'avenir" program, reference ANR-19-P3IA-0001 (PRAIRIE 3IA Institute). DC and GM thank Laurent Jacob, Louis Martin, François-Pierre Paty and Thomas Wolf for useful discussions.

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

# Appendix

Appendix A provides some background on notions used throughout the paper; Appendix B contains the proofs skipped in the paper; Appendix C provides additional experimental results as well as details on our protocol for reproducibility.

## A  ADDITIONAL BACKGROUND

This section provides some background on attention and transformers, Sinkhorn's algorithm and the relationship between optimal transport based kernels and positive definite histogram kernels.

### A.1  SINKHORN'S ALGORITHM: FAST COMPUTATION OF $\mathbf{P}_\kappa(\mathbf{x}, \mathbf{z})$

Without loss of generality, we consider here $\kappa$ the linear kernel. We recall that $\mathbf{P}_\kappa(\mathbf{x}, \mathbf{z})$ is the solution of an optimal transport problem, which can be efficiently solved by Sinkhorn's algorithm (Peyré & Cuturi, 2019) involving matrix multiplications only. Specifically, Sinkhorn's algorithm is an iterative matrix scaling method that takes the opposite of the pairwise similarity matrix $\mathbf{K}$ with entry $\mathbf{K}_{ij} := \langle \mathbf{x}_i, \mathbf{z}_j \rangle$ as input $\mathbf{C}$ and outputs the optimal transport plan $\mathbf{P}_\kappa(\mathbf{x}, \mathbf{z}) = \text{Sinkhorn}(\mathbf{K}, \varepsilon)$. Each iteration step $\ell$ performs the following updates

$$\mathbf{u}^{(\ell+1)} = \frac{1/n}{\mathbf{E}\mathbf{v}^{(\ell)}} \text{ and } \mathbf{v}^{(\ell+1)} = \frac{1/p}{\mathbf{E}^\top \mathbf{u}^{(\ell)}}, \tag{8}$$

where $\mathbf{E} = e^{\mathbf{K}/\varepsilon}$. Then the matrix $\text{diag}(\mathbf{u}^{(\ell)})\mathbf{E}\text{diag}(\mathbf{v}^{(\ell)})$ converges to $\mathbf{P}_\kappa(\mathbf{x}, \mathbf{z})$ when $\ell$ tends to $\infty$. However when $\varepsilon$ becomes too small, some of the elements of a matrix product $\mathbf{E}\mathbf{v}$ or $\mathbf{E}^\top \mathbf{u}$ become null and result in a division by 0. To overcome this numerical stability issue, computing the multipliers $\mathbf{u}$ and $\mathbf{v}$ is preferred (see *e.g.* (Peyré & Cuturi, 2019, Remark 4.23)). This algorithm can be easily adapted to a batch of data points $\mathbf{x}$, and with possibly varying lengths via a mask vector masking on the padding positions of each data point $\mathbf{x}$, leading to GPU-friendly computation. More importantly, all the operations above at each step are differentiable, which enables $\mathbf{z}$ to be optimized through back-propagation. Consequently, this module can be injected into any deep networks.

### A.2  ATTENTION AND TRANSFORMERS

We clarify the concept of attention — a mechanism yielding a context-dependent embedding for each element of $\mathbf{x}$ — as a special case of non-local operations (Wang et al., 2017; Buades et al., 2011), so that it is easier to understand its relationship to the OTK. Let us assume we are given a set $\mathbf{x} \in \mathcal{X}$ of length $n$. A non-local operation on $\mathbf{x}$ is a function $\Phi : \mathcal{X} \mapsto \mathcal{X}$ such that

$$\Phi(\mathbf{x})_i = \sum_{j=1}^n w(\mathbf{x}_i, \mathbf{x}_j)v(\mathbf{x}_j) = \mathbf{W}(\mathbf{x})_i^\top \mathbf{V}(\mathbf{x}),$$

where $\mathbf{W}(\mathbf{x})_i$ denotes the $i$-th column of $\mathbf{W}(\mathbf{x})$, a weighting function, and $\mathbf{V}(\mathbf{x}) = [v(\mathbf{x}_1), \dots, v(\mathbf{x}_n)]^\top$, an embedding. In contrast to operations on local neighborhood such as convolutions, non-local operations theoretically account for long range dependencies between elements in the set. In attention and the context of neural networks, $w$ is a *learned* function reflecting the *relevance* of each other elements $\mathbf{x}_j$ with respect to the element $\mathbf{x}_i$ being embedded and given the task at hand. In the context of the paper, we compare to a type of attention coined as *dot-product self-attention*, which can typically be found in the encoder part of the transformer architecture (Vaswani et al., 2017). Transformers are neural network models relying mostly on a succession of an attention layer followed by a fully-connected layer. Transformers can be used in sequence-to-sequence tasks — in this setting, they have an encoder with self-attention and a decoder part with a variant of self-attention —, or in sequence to label tasks, with only the encoder part. The paper deals with the latter. The name self-attention means that the attention is computed using a dot-product of linear

transformations of $\mathbf{x}_i$ and $\mathbf{x}_j$, and $\mathbf{x}$ attends to itself only. In its matrix formulation, dot-product self-attention is a non-local operation whose matching vector is

$$\mathbf{W}(\mathbf{x})_i = \text{Softmax}\left(\frac{W_Q\mathbf{x}_i\mathbf{x}^\top W_K^\top}{\sqrt{d_k}}\right),$$

where $W_Q \in \mathbb{R}^{n \times d_k}$ and $W_K \in \mathbb{R}^{n \times d_k}$ are learned by the network. In order to know which $\mathbf{x}_j$ are relevant to $\mathbf{x}_i$, the network computes scores between a query for $\mathbf{x}_i$ ($W_Q\mathbf{x}_i$) and keys of all the elements of $\mathbf{x}$ ($W_K\mathbf{x}$). The softmax turns the scores into a weight vector in the simplex. Moreover, a linear mapping $\mathbf{V}(\mathbf{x}) = W_V\mathbf{x}$, the values, is also learned. $W_Q$ and $W_K$ are often shared (Kitaev et al., 2020). A drawback of such attention is that for a sequence of length $n$, the model has to store an attention matrix $\mathbf{W}$ with size $O(n^2)$. More details can be found in Vaswani et al. (2017).

# B PROOFS

## B.1 PROOF OF LEMMA 3.1

*Proof.* First, since $\sum_{j=1}^{n'} p\mathbf{P}(\mathbf{x}', \mathbf{z})_{jk} = 1$ for any $k$, we have

$$W_2(\mathbf{x}, \mathbf{z})^2 = \sum_{i=1}^{n}\sum_{k=1}^{p}\mathbf{P}(\mathbf{x}, \mathbf{z})_{ik}d_\kappa^2(\mathbf{x}_i, \mathbf{z}_k)$$

$$= \sum_{i=1}^{n}\sum_{k=1}^{p}\sum_{j=1}^{n'}p\mathbf{P}(\mathbf{x}', \mathbf{z})_{jk}\mathbf{P}(\mathbf{x}, \mathbf{z})_{ik}d_\kappa^2(\mathbf{x}_i, \mathbf{z}_k)$$

$$= \|\mathbf{u}\|_2^2,$$

with $\mathbf{u}$ a vector in $\mathbb{R}^{nn'p}$ whose entries are $\sqrt{p\mathbf{P}(\mathbf{x}', \mathbf{z})_{jk}\mathbf{P}(\mathbf{x}, \mathbf{z})_{ik}}d_\kappa(\mathbf{x}_i, \mathbf{z}_k)$ for $i = 1, \ldots, n$, $j = 1, \ldots, n'$ and $k = 1, \ldots, p$. We can also rewrite $W_2^{\mathbf{z}}(\mathbf{x}, \mathbf{x}')$ as an $\ell_2$-norm of a vector $\mathbf{v}$ in $\mathbb{R}^{nn'p}$ whose entries are $\sqrt{p\mathbf{P}(\mathbf{x}', \mathbf{z})_{jk}\mathbf{P}(\mathbf{x}, \mathbf{z})_{ik}}d_\kappa(\mathbf{x}_i, \mathbf{x}'_j)$. Then by Minkowski inequality for the $\ell_2$-norm, we have

$$|W_2(\mathbf{x}, \mathbf{z}) - W_2^{\mathbf{z}}(\mathbf{x}, \mathbf{x}')| = |\|\mathbf{u}\|_2 - \|\mathbf{v}\|_2|$$

$$\leq \|\mathbf{u} - \mathbf{v}\|_2$$

$$= \left(\sum_{i=1}^{n}\sum_{k=1}^{p}\sum_{j=1}^{n'}p\mathbf{P}(\mathbf{x}', \mathbf{z})_{jk}\mathbf{P}(\mathbf{x}, \mathbf{z})_{ik}(d_\kappa(\mathbf{x}_i, \mathbf{z}_k) - d_\kappa(\mathbf{x}_i, \mathbf{x}'_j))^2\right)^{1/2}$$

$$\leq \left(\sum_{i=1}^{n}\sum_{k=1}^{p}\sum_{j=1}^{n'}p\mathbf{P}(\mathbf{x}', \mathbf{z})_{jk}\mathbf{P}(\mathbf{x}, \mathbf{z})_{ik}d_\kappa^2(\mathbf{x}'_j, \mathbf{z}_k)\right)^{1/2}$$

$$= \left(\sum_{k=1}^{p}\sum_{j=1}^{n'}\mathbf{P}(\mathbf{x}', \mathbf{z})_{jk}d_\kappa^2(\mathbf{x}'_j, \mathbf{z}_k)\right)^{1/2}$$

$$= W_2(\mathbf{x}', \mathbf{z}),$$

where the second inequality is the triangle inequality for the distance $d_\kappa$. Finally, we have

$$|W_2(\mathbf{x}, \mathbf{x}') - W_2^{\mathbf{z}}(\mathbf{x}, \mathbf{x}')|$$
$$\leq |W_2(\mathbf{x}, \mathbf{x}') - W_2(\mathbf{x}, \mathbf{z})| + |W_2(\mathbf{x}, \mathbf{z}) - W_2^{\mathbf{z}}(\mathbf{x}, \mathbf{x}')|$$
$$\leq W_2(\mathbf{x}', \mathbf{z}) + W_2(\mathbf{x}', \mathbf{z})$$
$$= 2W_2(\mathbf{x}', \mathbf{z}),$$

where the second inequality is the triangle inequality for the 2-Wasserstein distance. By symmetry, we also have $|W_2(\mathbf{x}, \mathbf{x}') - W_2^{\mathbf{z}}(\mathbf{x}, \mathbf{x}')| \leq 2W_2(\mathbf{x}', \mathbf{z})$, which concludes the proof. $\square$

## B.2 Relationship between $W_2$ and $W_2^{\mathbf{z}}$ for multiple references

Using the relation prooved in Appendix B.1, we can obtain a bound on the error term between $W_2$ and $W_2^{\mathbf{z}}$ for a data set of $m$ samples $(\mathbf{x}^1, \dots, \mathbf{x}^m)$ and $q$ references $(\mathbf{z}^1, \dots, \mathbf{z}^q)$

$$\mathcal{E}^2 := \frac{1}{m^2} \sum_{i,j=1}^m |W_2(\mathbf{x}^i, \mathbf{x}^j) - W_2^{\mathbf{z}^1, \dots, \mathbf{z}^q}(\mathbf{x}^i, \mathbf{x}^j)|^2 \leq \frac{4}{mq} \sum_{i=1}^m \sum_{j=1}^q W_2^2(\mathbf{x}^i, \mathbf{z}^j). \qquad (9)$$

When $q = 1$, the right-hand term in the inequality is the objective to minimize in the Wasserstein barycenter problem (Cuturi & Doucet, 2013), which further explains why we considered it: Once $W_2^{\mathbf{z}}$ is close to the Wasserstein distance $W_2$, $K_{\mathbf{z}}$ will also be close to $K_{\mathrm{OT}}$. We extend here the bound in equation 6 in the case of one reference to the multiple-reference case. The approximate 2-Wasserstein distance $W_2^{\mathbf{z}}(\mathbf{x}, \mathbf{x}')$ thus becomes

$$W_2^{\mathbf{z}^1, \dots, \mathbf{z}^q}(\mathbf{x}, \mathbf{x}') := \left\langle \frac{1}{q} \sum_{j=1}^q \mathbf{P}_{\mathbf{z}^j}(\mathbf{x}, \mathbf{x}'), d_\kappa^2(\mathbf{x}, \mathbf{x}') \right\rangle^{1/2} = \left( \frac{1}{q} \sum_{j=1}^q W_2^{\mathbf{z}^j}(\mathbf{x}, \mathbf{x}')^2 \right)^{1/2}.$$

Then by Minkowski inequality for the $\ell_2$-norm we have

$$|W_2(\mathbf{x}, \mathbf{x}') - W_2^{\mathbf{z}^1, \dots, \mathbf{z}^q}(\mathbf{x}, \mathbf{x}')| = \left| \left( \frac{1}{q} \sum_{j=1}^q W_2(\mathbf{x}, \mathbf{x}')^2 \right)^{1/2} - \left( \frac{1}{q} \sum_{j=1}^q W_2^{\mathbf{z}^j}(\mathbf{x}, \mathbf{x}')^2 \right)^{1/2} \right|$$

$$\leq \left( \frac{1}{q} \sum_{j=1}^q (W_2(\mathbf{x}, \mathbf{x}') - W_2^{\mathbf{z}^j}(\mathbf{x}, \mathbf{x}'))^2 \right)^{1/2},$$

and by equation 6 we have

$$|W_2(\mathbf{x}, \mathbf{x}') - W_2^{\mathbf{z}^1, \dots, \mathbf{z}^q}(\mathbf{x}, \mathbf{x}')| \leq \left( \frac{4}{q} \sum_{j=1}^q \min(W_2(\mathbf{x}, \mathbf{z}^j), W_2(\mathbf{x}', \mathbf{z}^j))^2 \right)^{1/2}.$$

Finally the approximation error in terms of Frobenius is bounded by

$$\mathcal{E}^2 := \frac{1}{m^2} \sum_{i,j=1}^m |W_2(\mathbf{x}^i, \mathbf{x}^j) - W_2^{\mathbf{z}^1, \dots, \mathbf{z}^q}(\mathbf{x}^i, \mathbf{x}^j)|^2 \leq \frac{4}{mq} \sum_{i=1}^m \sum_{j=1}^q W_2^2(\mathbf{x}^i, \mathbf{z}^j).$$

In particular, when $q = 1$ that is the case of single reference, we have

$$\mathcal{E}^2 \leq \frac{4}{m} \sum_{i=1}^m W_2^2(\mathbf{x}^i, \mathbf{z}),$$

where the right term equals to the objective of the Wasserstein barycenter problem, which justifies the choice of $\mathbf{z}$ when learning without supervision.

## C Additional Experiments and Setup Details

This section contains additional experiments on CIFAR-10, whose purpose is to illustrate the kernel associated with our embedding with respect to other classical or optimal transport based kernels, and test our embedding on another data modality; additional results for the experiments of the main section; details on our setup, in particular hyper-parameter tuning for our methods and the baselines.

### C.1 Experiments on Kernel Matrices (only for small data sets).

Here, we compare the optimal transport kernel $K_{\mathrm{OT}}$ (3) and its surrogate $K_{\mathbf{z}}$ (2) (with $\mathbf{z}$ learned without supervision) to common and other OT kernels. Although our embedding $\Phi_{\mathbf{z}}$ is scalable, the exact kernel require the computation of Gram matrices. For this toy experiment, we therefore use

Table 6: Classification accuracies for 5000 samples of CIFAR-10 using CKN features (Mairal, 2016) and forming Gram matrix. A random baseline would yield $10\%$.

| Dataset | $(3 \times 3)$, 256 | |
|---|---|---|
| Kernel | Accuracy | Runtime |
| Mean Pooling | 58.5 | $\sim 30$ sec |
| Flatten | 67.6 | $\sim 30$ sec |
| Sliced-Wasserstein (Kolouri et al., 2016) | 63.8 | $\sim 2$ min |
| Sliced-Wasserstein (Kolouri et al., 2016) + sin. pos enc. Devlin et al. (2019) | 66.0 | $\sim 2$ min |
| $K_{OT}$ | 64.5 | $\sim 20$ min |
| $K_{OT}$ + our pos enc. | 67.1 | $\sim 20$ min |
| $K_{\mathbf{z}}$ | 67.9 | $\sim 30$ sec |
| $K_{\mathbf{z}}$ + our pos enc. | 70.2 | $\sim 30$ sec |

5000 samples only of CIFAR-10 (images with $32 \times 32$ pixels), encoded without supervision using a two-layer convolutional kernel network (Mairal, 2016). The resulting features are $3 \times 3$ patches living in $\mathbb{R}^d$ with $d = 256$ or 8192. $K_{OT}$ and $K_{\mathbf{z}}$ aggregate existing features depending on the ground cost defined by $-\kappa$ (Gaussian kernel) given the computed weight matrix $\mathbf{P}$. In that sense, we can say that these kernels work as an adaptive pooling. We therefore compare it to kernel matrices corresponding to mean pooling and no pooling at all (linear kernel). We also compare to a recent positive definite and fast optimal transport based kernel, the Sliced Wasserstein Kernel (Kolouri et al., 2016) with 10, 100 and 1000 projection directions. We add a positional encoding to it so as to have a fair comparison with our kernels. A linear classifier is trained from this matrices. Although we cannot prove that $K_{OT}$ is positive definite, the classifier trained on the kernel matrix converges when $\varepsilon$ is not too small. The results can be seen on Table 6. Without positional information, our kernels do better than Mean pooling. When the positions are encoded, the Linear kernel is also outperformed. Note that including positions in Mean pooling and Linear kernel means interpolating between these two kernels: in the Linear kernel, only patches with same index are compared while in the Mean pooling, all patches are compared. All interpolations did worse than the Linear kernel. The runtimes illustrate the scalability of $K_{\mathbf{z}}$.

## C.2 CIFAR-10

Here, we test our embedding on the same data modality: we use CIFAR-10 features, *i.e.*, $60,000$ images with $32 \times 32$ pixels and 10 classes encoded using a two-layer CKN (Mairal, 2016), one of the baseline architectures for unsupervised learning of CIFAR-10, and evaluate on the standard test set. The very best configuration of the CKN yields a small number $(3 \times 3)$ of high-dimensional $(16, 384)$ patches and an accuracy of $85.8\%$. We will illustrate our embedding on a configuration which performs slightly less but provides more patches $(16 \times 16)$, a setting for which it was designed.

The input of our embedding are unsupervised features extracted from a 2-layer CKN with kernel sizes equal to 3 and 3, and Gaussian pooling size equal to 2 and 1. We consider the following configurations of the number of filters at each layer, to simulate two different input dimensions for our embedding:

- 64 filters at first and 256 at second layer, which yields a $16 \times 16 \times 256$ representation for each image.

- 256 filters at first and 1024 at second layer, which yields a $16 \times 16 \times 1024$ representation for each image.

Since the features are the output of a Gaussian embedding, $\kappa$ in our embedding will be the linear kernel. The embedding is learned with one reference and various supports using K-means method described in Section 3, and compared to several classical pooling baselines, including the original CKN's Gaussian pooling with pooling size equal to 6. The hyper-parameters are the entropic regularization $\varepsilon$ and bandwidth for position encoding $\sigma_{\text{pos}}$. Their search grids are shown in Table 7 and the results in Table 8. Without supervision, the adaptive pooling of the CKN features by our embedding notably improves their performance. We notice that the position encoding is very important to this task, which substantially improves the performance of its counterpart without it.

Table 7: Hyperparameter search range for CIFAR-10

| Hyperparameter | Search range |
|---|---|
| Entropic regularization $\varepsilon$ | $[1.0; 0.1; 0.01; 0.001]$ |
| Position encoding bandwidth $\sigma_{\text{pos}}$ | $[0.5; 0.6; 0.7; 0.8; 0.9; 1.0]$ |

Table 8: Classification results using unsupervised representations for CIFAR-10 for two feature configurations (extracted from a 2-layer unsipervised CKN with different number of filters). We consider here our embedding with one reference and different number of supports, learned with K-means, with or without position encoding (PE).

| Method | Nb. supports | $16 \times 16 \times 256$ | $16 \times 16 \times 1024$ |
|---|---|---|---|
| Flatten | | 73.1 | 76.1 |
| Mean pooling | | 64.9 | 73.4 |
| Gaussian pooling (Mairal, 2016) | | 77.5 | 82.0 |
| Ours | 9 | 75.6 | 79.3 |
| Ours (with PE) | | 78.0 | 82.2 |
| Ours | 64 | 77.9 | 80.1 |
| Ours (with PE) | | 81.4 | 83.2 |
| Ours | 144 | 78.4 | 80.7 |
| Ours (with PE) | | 81.8 | 83.4 |

## C.3 PROTEIN FOLD RECOGNITION

**Dataset description.** Our protein fold recognition experiments consider the Structural Classification Of Proteins (SCOP) version 1.75 and 2.06. We follow the data preprocessing protocols in Hou et al. (2019), which yields a training and validation set composed of 14699 and 2013 sequences from SCOP 1.75, and a test set of 2533 sequences from SCOP 2.06. The resulting protein sequences belong to 1195 different folds, thus the problem is formulated as a multi-classification task. The input sequence is represented as a 45-dimensional vector at each amino acid. The vector consists of a 20-dimensional one-hot encoding of the sequence, a 20-dimensional position-specific scoring matrix (PSSM) representing the profile of amino acids, a 3-class secondary structure represented by a one-hot vector and a 2-class solvent accessibility. The lengths of the sequences are varying from tens to thousands.

**Models setting and hyperparameters.** We consider here the one-layer models followed by a global mean pooling for the baseline methods CKN (Chen et al., 2019a) and RKN (Chen et al., 2019b). We build our model on top of the one-layer CKN model, where $\kappa$ can be seen as a Gaussian kernel on the k-mers in sequences. The only difference between our model and CKN is thus the pooling operation, which is given by our embedding introduced in Section 3. The bandwidth parameter of the Gaussian kernel $\kappa$ on k-mers is fixed to 0.6 for unsupervised models and 0.5 for supervised models, the same as used in CKN which were selected by the accuracy on the validation set. The filter size $k$ is fixed to 10 and different numbers of anchor points in Nyström for $\kappa$ are considered in the experiments. The other hyperparameters for our embedding are the entropic regularization parameter $\varepsilon$, the number of supports in a reference $p$, the number of references $q$, the number of iterations for Sinkhorn's algorithm and the regularization parameter $\lambda$ in the linear classifier. The search grid for $\varepsilon$ and $\lambda$ is shown in Table 9 and they are selected by the accuracy on validation set. $\varepsilon$ plays an important role in the performance and is observed to be stable for the same dataset. For this dataset, it is selected to be 0.5 for all the unsupervised and supervised models. The effect of other hyperparameters will be discussed below.

For the baseline methods, the accuracies of PSI-BLAST and DeepSF are taken from Hou et al. (2019). The hyperparameters for CKN and RKN can be found in Chen et al. (2019b). For Rep the Set (Skianis et al., 2020) and Set Transformer (Lee et al., 2019), we use the public implementations by the authors. These two models are used on the top of a convolutional layer of the same filter size as CKN to extract $k$-mer features. As the exact version of Rep the Set does not provide any implementation for back-propagation to a bottom layer of it, we consider the approximate version of

Table 9: Hyperparameter search grid for SCOP 1.75

| Hyperparameter | Search range |
|---|---|
| $\varepsilon$ for Sinkhorn | $[1.0; 0.5; 0.1; 0.05; 0.01]$ |
| $\lambda$ for classifier (unsupervised setting) | $1/2^{\text{range}(5,20)}$ |
| $\lambda$ for classifier (supervised setting) | $[1e\text{-}6; 1e\text{-}5; 1e\text{-}4; 1e\text{-}3]$ |

Table 10: Hyperparameter search grid for SCOP 1.75 baselines.

| Model and Hyperparameter | Search range |
|---|---|
| ApproxRepSet: Hidden Sets $\times$ Cardinality | $[20; 30; 50; 100] \times [10; 20; 50]$ |
| ApproxRepSet: Learning Rate | $[0.0001; 0.0005; 0.001]$ |
| ApproxRepSet: Weight Decay | $[1e\text{-}5; 1e\text{-}4; 1e\text{-}3; 1e\text{-}2]$ |
| Set Transformer: Heads $\times$ Dim Hidden | $[1; 4; 8] \times [64; 128; 256]$ |
| Set Transformer: Learning Rate | $[0.0001; 0.0005; 0.001]$ |
| Set Transformer: Weight Decay | $[1e\text{-}5; 1e\text{-}4; 1e\text{-}3; 1e\text{-}2]$ |

Rep the Set only, which also scales better to our dataset. The default architecture of Set Transformer did not perform well due to overfitting. We therefore used a shallower architecture with one ISAB, one PMA and one linear layer, similar to the one-layer architectures of CKN and our model. We tuned their model hyperparameters, weight decay and learning rate. The search grids for these hyperparameters are shown in Table 10.

**Unsupervised embedding.** The kernel embedding $\varphi$, which is infinite dimensional for the Gaussian kernel, is approximated with the Nyström method using K-means on 300000 k-mers extracted from the same training set as in Chen et al. (2019b). The reference measures are learned by using either K-means or Wasserstein to update centroids in 2-Wasserstein K-means on 3000 subsampled sequences for RAM-saving reason. We evaluate our model on top of features extracted from CKNs of different dimensions, representing the number of anchor points used to approximate $\kappa$. The number of iterations for Sinkhorn is fixed to 100 to ensure the convergence. The results for different combinations of $q$ and $p$ are provided in Table 11. Increasing the number of supports $p$ can improve the performance and then saturate it when $p$ is too large. On the other hand, increasing the number of references while keeping the embedding dimension (*i.e.* $p \times q$) constant is not significantly helpful in this unsupervised setting. We also notice that Wasserstein Barycenter for learning the references does not outperform K-means, while the latter is faster in terms of computation.

**Supervised embedding.** Our supervised embedding is initialized with the unsupervised method and then trained in an alternating fashion which was also used for CKN: we use an Adam algorithm to update anchor points in Nyström and reference measures $\mathbf{z}$, and the L-BFGS algorithm to optimize

Table 11: Classification accuracy (top 1/5/10) results of our unsupervised embedding for SCOP 1.75. We show the results for different combinations of (number of references $q \times$ number of supports $p$). The reference measures $\mathbf{z}$ are learned with either K-means or Wasserstein barycenter for updating centroids.

| Nb. filters | Method | $q$ | Embedding size ($q \times p$) | | | |
|---|---|---|---|---|---|---|
| | | | 10 | 50 | 100 | 200 |
| 128 | K-means | 1 | 76.5/91.5/94.4 | 77.5/91.7/94.5 | 79.4/92.4/94.9 | 78.7/92.1/95.1 |
| | | 5 | 72.8/89.9/93.7 | 77.8/91.7/94.6 | 78.6/91.9/94.6 | 78.1/92.1/94.7 |
| | | 10 | 62.7/85.8/91.1 | 76.5/91.0/94.2 | 78.1/92.2/94.9 | 78.6/92.2/94.7 |
| | Wass. bary. | 1 | 64.0/85.9/91.5 | 71.6/88.9/93.2 | 77.2/91.4/94.2 | 77.5/91.9/94.8 |
| | | 5 | 70.5/89.1/93.0 | 76.6/91.3/94.4 | 78.4/91.7/94.3 | 77.1/91.9/94.7 |
| | | 10 | 63.0/85.7/91.0 | 75.9/91.4/94.3 | 77.5/91.9/94.6 | 77.7/92.0/94.7 |
| 1024 | K-means | 1 | 84.4/95.0/96.6 | 84.6/95.0/97.0 | 85.7/95.3/96.7 | 85.4/95.2/96.7 |
| | | 5 | 81.1/94.0/96.2 | 84.9/94.8/96.8 | 84.7/94.4/96.7 | 85.2/95.0/96.7 |
| | | 10 | 79.8/93.5/95.9 | 83.1/94.6/96.6 | 84.4/94.7/96.7 | 84.8/94.9/96.7 |

Table 12: Classification accuracy (top 1/5/10) results of our unsupervised embedding for SCOP 1.75. We show the results for different number of Nyström anchors. The number of references and supports are fixed to 1 and 100.

| Number of anchors | Accuracies |
|---|---|
| 1024 | 85.8/95.3/96.8 |
| 2048 | 86.6/95.9/97.2 |
| 3072 | 87.8/96.1/97.4 |

Table 13: Classification accuracy (top 1/5/10) of supervised models for SCOP 1.75. The accuracies obtained by averaging 10 different runs. We show the results of using either one reference with 50 supports or 5 references with 10 supports. Here DeepSF is a 10-layer CNN model.

| Method | Runtime | Top 1/5/10 accuracy on SCOP 2.06 | |
|---|---|---|---|
| PSI-BLAST (Hou et al., 2019) | - | 84.53/86.48/87.34 | |
| DeepSF (Hou et al., 2019) | - | 73.00/90.25/94.51 | |
| Set Transformer (Lee et al., 2019) | 3.3h | 79.15±4.61/91.54±1.40/94.33±0.63 | |
| ApproxRepSet (Skianis et al., 2020) | 2h | 84.51±0.58/94.03±0.44/95.73±0.37 | |
| Number of filters | | 128 | 512 |
| CKN (Chen et al., 2019a) | 1.5h | 76.30±0.70/92.17±0.16/95.27±0.17 | 84.11±0.11/94.29±0.20/96.36±0.13 |
| RKN (Chen et al., 2019b) | - | 77.82±0.35/92.89±0.19/95.51±0.20 | 85.29±0.27/94.95±0.15/96.54±0.12 |
| Ours | | | |
| $\Phi_{\mathbf{z}}$ (1 × 50) | 3.5h | 82.83±0.41/93.89±0.33/96.23±0.12 | 88.40±0.22/95.76±0.13/97.10±0.15 |
| $\Phi_{\mathbf{z}}$ (5 × 10) | 4h | **84.68±0.50/94.68±0.29/96.49±0.18** | **88.66±0.25/95.90±0.15/97.33±0.14** |

the classifier. The learning rate for Adam is initialized with 0.01 and halved as long as there is no decrease of the validation loss for 5 successive epochs. In practice, we notice that using a small number of Sinkhorn iterations can achieve similar performance to a large number of iteration, while being much faster to compute. We thus fix it to 10 throughout the experiments. The accuracy results are obtained by averaging on 10 runs with different seeds following the setting in Chen et al. (2019b). The results are shown in Table 13 with error bars. The effect of the number of supports $q$ is similar to the unsupervised case, while increasing the number of references can indeed improve performance.

## C.4 DETECTION OF CHROMATIN PROFILES

**Dataset description.** Predicting the functional effects of noncoding variants from only genomic sequences is a central task in human genetics. A fundamental step for this task is to simultaneously predict large-scale chromatin features from DNA sequences (Zhou & Troyanskaya, 2015). We consider here the DeepSEA dataset, which consists in simultaneously predicting 919 chromatin profiles including 690 transcription factor (TF) binding profiles for 160 different TFs, 125 DNase I sensitivity profiles and 104 histone-mark profiles. In total, there are 4.4 million, 8000 and 455024 samples for training, validation and test. Each sample consists of a 1000-bp DNA sequence from the human GRCh37 reference. Each sequence is represented as a $1000 \times 4$ binary matrix using one-hot encoding on DNA characters. The dataset is available at `http://deepsea.princeton.edu/media/code/deepsea_train_bundle.v0.9.tar.gz`. Note that the labels for each profile are very imbalanced in this task with few positive samples. For this reason, learning unsu-

Table 14: Classification accuracy (top 1/5/10) results of our supervised embedding for SCOP 1.75. We show the results for different combinations of (number of references $q$ × number of supports $p$). The reference measures $\mathbf{z}$ are learned with K-means.

| Embedding size ($q \times p$) | 10 | 50 | 100 | 200 |
|---|---|---|---|---|
| $q = 1$ | 88.3/95.5/97.0 | 88.4/95.8/97.2 | 87.1/94.9/96.7 | 87.7/94.9/96.3 |
| $q = 2$ | 87.8/95.8/97.0 | 89.6/96.2/97.5 | 86.5/94.9/96.6 | 87.6/94.9/96.3 |
| $q = 5$ | 87.0/95.1/96.7 | 88.8/96.0/97.2 | 87.4/95.4/97.0 | 87.4/94.7/96.2 |
| $q = 10$ | 84.5/93.6/95.6 | 89.8/96.0/97.2 | 88.0/95.7/97.0 | 85.6/94.4/96.1 |

Table 15: Model architecture for DeepSEA dataset.

| Model architecture |
| --- |
| Conv1d(in channels=4, out channels=$d$, kernel size=16) + ReLU |
| (Ours) EmbeddingLayer(in channels=$d$, supports=64, references=1, $\varepsilon = 1.0$, PE=True, $\sigma_{\text{pos}} = 0.1$) |
| Linear(in channels=$d$, out channels=$d$) + ReLU |
| Dropout(0.4) |
| Linear(in channels=$d \times 64$, out channels=919) + ReLU |
| Linear(in channels=919, out channels=919) |

Table 16: Results for prediction of chromatin profiles on the DeepSEA dataset. The metrics are area under ROC (auROC) and area under PR curve (auPRC), averaged over 919 chromatin profiles. The accuracies are averaged from 10 different runs. Armed with the positional encoding (PE) described in Section 3, our embedding outperforms the state-of-the-art model and another model of our embedding with the PE proposed in Vaswani et al. (2017).

| Method | DeepSEA | Ours | Ours ($d = 1024$) | Ours ($d = 1536$) |
| --- | --- | --- | --- | --- |
| Position encoding | - | Sinusoidal (Vaswani et al., 2017) | Ours | Ours |
| auROC | 0.933 | 0.917 | 0.935 | **0.936** |
| auPRC | 0.342 | 0.311 | 0.354 | **0.360** |

pervised models could be intractable as they may require an extremely large number of parameters if junk or redundant sequences cannot be filtered out.

**Model architecture and hyperparameters.** For the above reason and fair comparison, we use here our supervised embedding as a module in Deep NNs. The architecture of our model is shown in Table 15. We use an Adam optimizer with initial learning rate equal to 0.01 and halved at epoch 1, 4, 8 for 15 epochs in total. The number of iterations for Sinkhorn is fixed to 30. The whole training process takes about 30 hours on a single GTX2080TI GPU. The dropout rate is selected to be 0.4 from the grid $[0.1; 0.2; 0.3; 0.4; 0.5]$ and the weight decay is 1e-06, the same as Zhou & Troyanskaya (2015). The $\sigma_{\text{pos}}$ for position encoding is selected to be 0.1, by the validation accuracy on the grid $[0.05; 0.1; 0.2; 0.3; 0.4; 0.5]$. The checkpoint with the best validation accuracy is used to evaluate on the test set. Area under ROC (auROC) and area under precision curve (auPRC), averaged over 919 chromatin profiles, are used to measure the performance. The hidden size $d$ is chosen to be either 1024 or 1536.

**Results and importance of position encoding.** We compare our model to the state-of-the-art CNN model DeepSEA (Zhou & Troyanskaya, 2015) with 3 convolutional layers, whose best hyperparameters can be found in the corresponding paper. Our model outperforms DeepSEA, while requiring fewer layers. The positional information is known to be important in this task. To show the efficacy of our position encoding, we compare it to the sinusoidal encoding used in the original transformer (Vaswani et al., 2017). We observe that our encoding with properly tuned $\sigma_{\text{pos}}$ requires fewer layers, while being interpretable from a kernel point of view. We also find that larger hidden size $d$ performs better, as shown in Table 16. ROC and PR curves for all the chromatin profiles and stratified by transcription factors, DNase I-hypersensitive sites and histone-marks can also be found in Figure 2.

## C.5 SST-2

**Dataset description.** The data set contains 67,349 training samples and 872 validation samples and can be found at `https://gluebenchmark.com/tasks`. The test set contains 1,821 samples for which the predictions need to be submitted on the GLUE leaderboard, with limited number of submissions. As a consequence, our training and validation set are extracted from the original training set ($80\%$ of the original training set is used for our training set and the remaining $20\%$ is used for our validation set), and we report accuracies on the standard validation set, used as a test set. The reviews are padded with zeros when their length is shorter than the chosen sequence length (we choose 30 and 66, the latter being the maximum review length in the data set) and the BERT

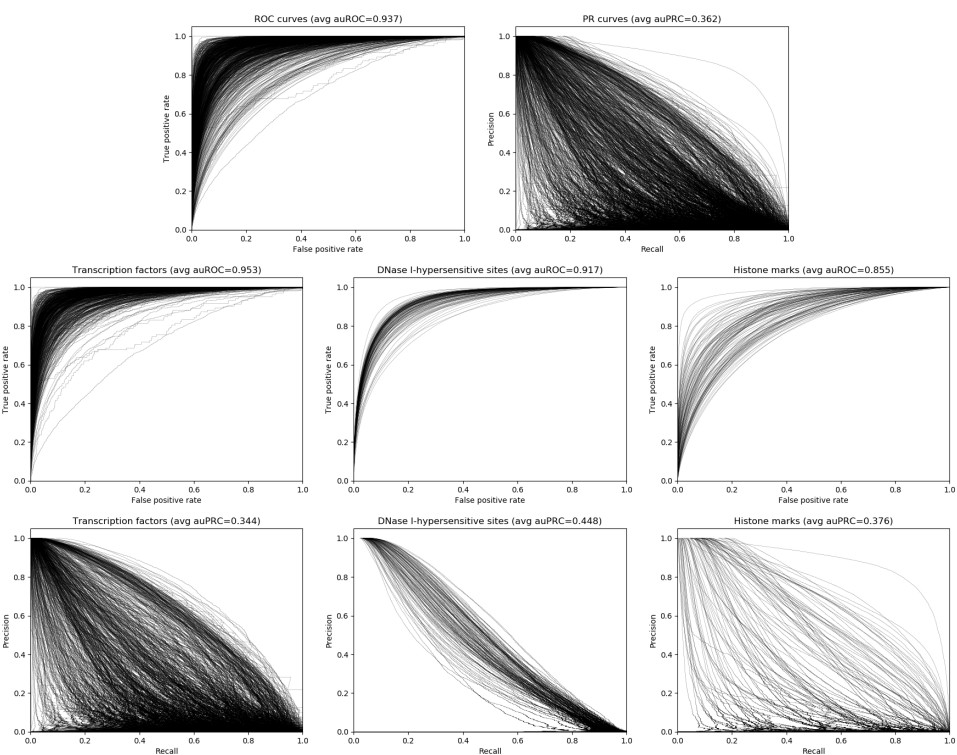

Figure 2: ROC and PR curves for all the chromatin profiles (first row) and stratified by transcription factors (left column), DNase I-hypersensitive sites (middle column) and histone-marks (right column). The profiles with positive samples fewer than 50 on the test set are not taken into account.

Table 17: Accuracies on standard validation set for SST-2 with our unsupervised features depending on the number of references and supports. The references were computed using K-means on samples for multiple references and K-means on patches for multiple supports. The size of the input BERT features is (length $\times$ dimension). The accuracies are averaged from 10 different runs. ($q$ references $\times$ $p$ supports)

| BERT Input Feature Size | $(30 \times 768)$ | | $(66 \times 768)$ | |
|---|---|---|---|---|
| Features | Pre-trained | Fine-tuned | Pre-trained | Fine-tuned |
| [CLS] | 84.6±0.3 | 90.3±0.1 | 86.0±0.2 | **92.8±0.1** |
| Flatten | 84.9±0.4 | **91.0±0.1** | 85.2±0.3 | 92.5±0.1 |
| Mean pooling | 85.3±0.3 | 90.8±0.1 | 85.4±0.3 | 92.6±0.2 |
| $\Phi_\mathbf{z}$ $(1 \times 3)$ | 85.5±0.1 | 90.9±0.1 | 86.5±0.1 | 92.6±0.1 |
| $\Phi_\mathbf{z}$ $(1 \times 10)$ | 85.1±0.4 | 90.9±0.1 | 85.9±0.3 | 92.6±0.1 |
| $\Phi_\mathbf{z}$ $(1 \times 30)$ | 86.3±0.3 | 90.8±0.1 | 86.6±0.5 | 92.6±0.1 |
| $\Phi_\mathbf{z}$ $(1 \times 100)$ | 85.7±0.7 | 90.9±0.1 | 86.6±0.1 | 92.7±0.1 |
| $\Phi_\mathbf{z}$ $(1 \times 300)$ | **86.8±0.3** | 90.9±0.1 | **87.2±0.1** | 92.7±0.1 |

Table 18: Hyperparameter search grid for SST-2.

| Hyperparameter | Search range |
|---|---|
| Entropic regularization $\varepsilon$ | $[0.5; 1.0; 3.0; 10.0]$ |
| $\lambda$ for classifier (unsupervised setting) | $10^{\text{range}(-10,1)}$ |
| Gaussian kernel bandwidth | $[0.4; 0.5; 0.6; 0.7; 0.8]$ |
| Learning rate (supervised setting) | $[0.1; 0.01; 0.001]$ |

implementation requires to add special tokens [CLS] and [SEP] at the beginning and the end of each review.

**Model architecture and hyperparameters.** In most transformers such as BERT, the embedding associated to the token [CLS] is used for classification and can be seen in some sense as an embedding of the review adapted to the task. The features we used are the word features provided by the BERT base-uncased version, available at `https://huggingface.co/transformers/pretrained_models.html`. For this version, the dimension of the word features is 768. Our model is one layer of our embedding, with $\varphi$ the Gaussian kernel mapping with varying number of Nyström filters in the supervised setting, and the Linear kernel in the unsupervised setting. We do not add positonnal encoding as it is already integrated in BERT features. In the unsupervised setting, the output features are used to train a large-scale linear classifier, a Pytorch linear layer. We choose the best hyper-parameters based on the accuracy of a validation set. In the supervised case, the parameters of the previous model, $\mathbf{w}$ and $\mathbf{z}$, are optimized end-to-end. In this case, we tune the learning rate. In both case, we tune the entropic regularization parameter of optimal transport and the bandwidth of the Gaussian kernel. The parameters in the search grid are summed up in Table 18. The best entropic regularization and Gaussian kernel bandwidth are typically and respectively 3.0 and 0.5 for this data set. The supervised training process takes between half an hour for smaller models (typically 128 filters in $\mathbf{w}$ and 3 supports in $\mathbf{z}$) and a few hours for larger models (256 filters and 100 supports) on a single GTX2080TI GPU. The hyper-parameters of the baselines were similarly tuned, see 19. Mean Pooling and [CLS] embedding did not require any tuning except for the regularization $\lambda$ of the classifier, which followed the same grid as in Table 18.

**Results and discussion.** As explained in Section 5, our unsupervised embedding improves the BERT pre-trained features while still using a simple linear model as shown in Table 17, and its supervised counterpart enables to get even closer to the state-of-the art (for the BERT base-uncased model) accuracy, which is usually obtained after fine-tuning of the BERT model on the whole data set. This can be seen in Tables 20; 21. We also add a baseline consisting of one layer of classical self-attention, which did not do well hence was not reported in the main text.

Table 19: Hyperparameter search grid for SST-2 baselines.

| Model and Hyperparameter | Search range |
|---|---|
| RepSet and ApproxRepSet: Hidden Sets $\times$ Cardinality | $[4; 20; 30; 50; 100] \times [3; 10; 20; 30; 50]$ |
| ApproxRepSet: Learning Rate | $[0.0001; 0.001; 0.01]$ |
| Set Transformer: Heads $\times$ Dim Hidden | $[1; 4] \times [8; 16; 64; 128]$ |
| Set Transformer: Learning Rate | $[0.001; 0.01]$ |

Table 20: Classification accuracy on standard validation set of supervised models for SST-2, with pre-trained BERT ($30 \times 768$) features. The accuracies of our embedding were averaged from 3 different runs before being run 10 times for the best results for comparison with baselines, cf. Section 5. 10 Sinkhorn iterations were used. ($q$ references $\times$ $p$ supports).

| Method | Accuracy on SST-2 | | |
|---|---|---|---|
| Number of Nyström filters | 32 | 64 | 128 |
| $\Phi_{\mathbf{z}}$ $(1 \times 3)$ | 88.38 | 88.38 | 88.18 |
| $\Phi_{\mathbf{z}}$ $(1 \times 10)$ | 88.11 | 88.15 | 87.61 |
| $\Phi_{\mathbf{z}}$ $(1 \times 30)$ | 88.30 | 88.30 | 88.26 |
| $\Phi_{\mathbf{z}}$ $(4 \times 3)$ | 88.07 | 88.26 | 88.30 |
| $\Phi_{\mathbf{z}}$ $(4 \times 10)$ | 87.6 | 87.84 | 88.11 |
| $\Phi_{\mathbf{z}}$ $(4 \times 30)$ | 88.18 | **88.68** | 88.07 |

Table 21: Classification accuracy on standard validation set of all baselines for SST-2, with pre-trained BERT ($30 \times 768$) features, averaged from 10 different runs.

| Method | Accuracy on SST-2 |
|---|---|
| [CLS] embedding (Devlin et al., 2019) | 90.3$\pm$0.1 |
| Mean Pooling of BERT features (Devlin et al., 2019) | **90.8 $\pm$ 0.1** |
| One Self-Attention Layer (Vaswani et al., 2017) | 83.7$\pm$0.1 |
| Approximate Rep the Set (Skianis et al., 2020) | 86.8$\pm$0.9 |
| Rep the Set (Skianis et al., 2020) | 87.1$\pm$0.5 |
| Set Transformer (Lee et al., 2019) | 87.9$\pm$0.8 |
| $\Phi_{\mathbf{z}}$ $(1 \times 30)$ (dot-product instead of OT) | 86.9$\pm$1.1 |

