# OpenReview forum: "A Trainable Optimal Transport Embedding for Feature Aggregation and its Relationship to Attention"
_ICLR.cc/2021/Conference — ICLR 2021 Poster_

### Official Review · AnonReviewer2 · 2020-10-28
**Interesting idea**

**Rating:** 7
**Confidence:** 2

**Review:**

##########################################################################

Summary:

The paper proposes a parametrized embedding based on optimal transport in the kernel space. The method is scalable and the parameters of the embedding can be learned in either unsupervised or supervised fashion. The paper demonstrates the effectiveness of the purposed approach using real data in bioinformatics and natural language processing.

Overall, the proposed embedding seems to be novel and well-motivated. The empirical results are impressive. Also, the paper is well written and technically sound. I don't have major concerns about the paper.


##########################################################################

Minor comments:

- Contributions paragraph: "in either unsupervised and supervised fashion" --> "in either unsupervised or supervised fashion"
- The line below Eq. (1): the entropy should be - \sum P ( log P - 1 ) ?
- Section 3.2 "Infinite-dimensional embedding in RKHS" line 1: "a reference z in X" repeated twice.
- Third line below Eq. (4): \Phi_z(x) instead of \Phi_z(z)
- First line in Section 5.2: "either unsupervised or supervised settings" --> "both unsupervised and supervised settings"

---

> ### Author Response · Authors · 2020-11-17
> **individual response to R2**
>
> We thank the reviewer for his/her feedback and for his/her encouraging comments. We are currently working on a revision of the paper, which will take into account the reported typos. In particular, the entropic penalty is indeed  - \sum P ( log P - 1 ) and not - \sum P log (P - 1).

---

### Official Review · AnonReviewer3 · 2020-10-29
**Simple, but reasonable approach**

**Rating:** 6
**Confidence:** 2

**Review:**

The paper proposes a transport-based feature representation for the input of a set of vectors. The feature is defined through the transport to reference sets. Vectors in reference sets can be learned by supervised or unsupervised approaches. The relation with self-attention is also discussed. Experiments empirically show a good performance of the proposed method.

Transporting a set of inputs to trainable references is seemingly a more sophisticated extension of the classical basis function regression with trainable basis (basis corresponds to references), and thus, I think the basic idea would be reasonable and easy to have an intuition. My major concern is on about the settings of the number of references and supports, which would have a large number of possible settings. Except for that, I currently only have minor comments.

Discussing the selection of the number of references and supports in more detail would have been informative. In my understanding, these are most important hyper-parameters. Currently, no practical recipes and analysis of sensitivity to these settings are provided. How were they chosen in the experiments?

The number of supports can be different for each one of references. Therefore, there exist a huge number of possible settings even when only considering moderate sizes.

The optimized references seemingly have some interpretation about discovery of an important set of vectors. I'm not fully sure, but it seems to provide a different view point from self-attention, and thus, it is perhaps worth mentioning if possible.

A naive extension could be to make multiple transportation layers (transport z to another references). Is it meaningless? (I guess, at least, some activation function is required) If it has some meaning, providing comments may be informative.

In eq.(5), 'R^k x p' should be 'R^p x k'?

---

> ### Author Response · Authors · 2020-11-17
> **individual response to R3**
>
> > Major concern: settings of the number of references and supports, which would have a large number of possible settings. Discussing the selection of the number of references and supports in more detail would have been informative. In my understanding, these are most important hyper-parameters. Currently, no practical recipes and analysis of sensitivity to these settings are provided. How were they chosen in the experiments? The number of supports can be different for each one of references. Therefore, there exist a huge number of possible settings even when only considering moderate sizes.
>
>  This is indeed an important aspect that we will clarify in the paper. In particular, each reference has the same number of supports in our implementation for the sake of simplicity and similarity to the transformer, but it would be interesting to investigate the possibility of varying the number of supports. Regarding the choice of parameters, we have addressed this point in the general comments. Note that each configuration choice was made by optimizing the validation error.
>
> > The optimized references seemingly have some interpretation about discovery of an important set of vectors. I'm not fully sure, but it seems to provide a different view point from self-attention, and thus, it is perhaps worth mentioning if possible.
>
> We agree with the reviewer and propose to emphasize this point in Section 4. Visualization and interpretation of the learned references would be of interest for biological sequences, which we plan to investigate for future work. We thank the reviewer for this suggestion.
>
> > A naive extension could be to make multiple transportation layers (transport z to another references). Is it meaningless? (I guess, at least, some activation function is required) If it has some meaning, providing comments may be informative.
>
> This is an interesting question and wether it is meaningless or not is not clear to us yet. We discuss this point in the general comments.
>
> > In eq.(5), 'R^k x p' should be 'R^p x k'?
>
> This was indeed a typo. Thank you for noting this.

---

### Official Review · AnonReviewer1 · 2020-10-29
**Nice and well-executed idea for feature aggregation based on OT (with connections to attention)**

**Rating:** 7
**Confidence:** 4

**Review:**

**Summary**:
The authors propose a new way to aggregate the embeddings of elements in a set (or sequence) by comparing it with respect to (trainable) reference set(s) via Optimal Transport (OT). The motivation to build such a pooling operation is derived from self-attention and the authors suggest an OT spin to that (e.g., the different reference sets/measures can be thought of as different heads in attention). This is, however, done in a principled way with the help of kernel embeddings and not just ad-hoc using the transport plan as the attention matrix.

**Pros**:
-They properly bridge the gap of OT with attention via their non-local pooling perspective. This allows them to have a similar feature aggregator, but which requires linear memory as compared to quadratic for the vanilla self-attention (although there have been several recent linear memory variants to it, any comments?).
-The method results in an improvement over other baselines for several biology-based sequencing applications (also they demonstrate early results for NLP).
-The paper is well written and clear.

**Cons**:
-The empirical results are okay, but could have been better or tested more extensively. Also, there are some peculiarities about their empirical analysis.
-This kind of technique has promise for use in NLP, but I am slightly worried that right now there are just too many "knobs" whose correct configuration will have to be worked out (anchor points, reference measures, nystrom, etc). However, I think this can be left to future work.

*-> It would be great if the authors can answer the questions below:*

Empirical analysis:
-In the experiments for Table 2, 3, 4, do you use the same number of heads for other baselines as the number of references in your case?

-It seems quite funny that, for their unsupervised results, the optimal number of supports p equal 1, which would basically imply that optimal transport wasn’t much of use and the features of x_i were just pooled based on an average. Do you have any comments on this and why this is happening?

-The NLP experiments are relatively weak and could still have been strengthened by testing on more tasks from the GLUE benchmark. Another surprising thing is that even though scores of your method unsupervised are higher than mean pooling of BERT features, the latter significantly outperforms in the supervised case. Do you have any explanation for this?

-Besides, why do you write in the text that NLP is “an a priori less favourable setting”?

-Can you give some precise estimates of how the runtime of this OTKE compares to dot product attention? Also, how expensive is the Nystrom procedure?

Miscellaneous:
-This is a bit of nitpicking, but the authors are a bit lazy while citing, and in several places just cite the textbooks for OT and Kernels (which is not a problem). However, it is useful for the reader to additionally have the accurate references as well, e.g. for Sinkhorn’s algorithm perhaps also cite Sinkhorn & Knopp (1967) and Cuturi (2013); for OT theory, Villani 2008 etc.
-Typos: "a set x and a reference z in X and a reference z”
-ISAB and PMA (unexplained acronyms)

**Conclusion**:  Overall, I think the idea is quite interesting, and carries the potential for OT-based building blocks in NLP, or even analysing the benefit provided by attention. Hence, I am in favour of accepting this paper.

(PS: I might be slightly biased because I like OT and have also been thinking about ideas on similar lines connecting attention and OT.)

---

> ### Author Response · Authors · 2020-11-17
> **individual response to R1**
>
> We thank the reviewer for his/her detailed feedback.
>
>
> > There have been several recent linear memory variants to it (attention), any comments?
>
> We agree with the reviewer that a discussion is missing. We have addressed this point in the general comment section above.
>
> > I am slightly worried that right now there are just too many "knobs" whose correct configuration will have to be worked out (anchor points, reference measures, nystrom, etc). However, I think this can be left to future work.
>
> We believe that our approach and code are very easy to use (after all, the number of parameters of the OTKE layer is similar to that of a two-layer neural network), but that indeed a summary of the impact of these parameters is missing in the paper. We have addressed this point in the general comments.
>
> > Empirical analysis: in the experiments for Table 2, 3, 4, do you use the same number of heads for other baselines as the number of references in your case?
>
> - Table 2;4: We tuned the number of heads and hidden dimensions for other baselines following the grids used in the original papers (which are relatively consistent with our grids for OT). The search grids for these hyper-parameters are detailed in Table 10 and 18.
> - Table 3: we used one reference in our model. The baseline (and state-of-the-art) architecture for this task is a CNN hence does not have attention heads.
>
> > Empirical analysis: it seems quite funny that, for their unsupervised results, the optimal number of supports p equal 1, which would basically imply that optimal transport wasn’t much of use and the features of x_i were just pooled based on an average. Do you have any comments on this and why this is happening?
>
> The answer to this question is very simple: **we made a typo!** The legend should be "q references x p supports" instead of "p supports x q references". We thank you for noting this very confusing point.
>
> > Empirical analysis: The NLP experiments are relatively weak and could still have been strengthened by testing on more tasks from the GLUE benchmark. Another surprising thing is that even though scores of your method unsupervised are higher than mean pooling of BERT features, the latter significantly outperforms in the supervised case. Do you have any explanation for this?
>
> In the supervised case, all parameters of BERT are fine-tuned on the SST-2 data set while the OTKE is trained end-to-end on the frozen BERT features without fine-tuning. Moreover, we believe many NLP tasks to have different characteristics from problems for which our embedding was initially designed (see below and also response to R4).
>
> > Besides, why do you write in the text that NLP is “an a priori less favourable setting”?
>
> Our work was mainly motivated by long sequences with few labeled data, where its competitive advantage lies for now. In most GLUE tasks, the sequences are shorter (e.g, <66 words for SST-2) and/or there are more labeled data per class than in our bioinformatics tasks. We therefore expected the gain with our method to be less substantial than other baselines designed and benchmarked on NLP GLUE tasks, e.g, fully-trained BERT.
>
> > Can you give some precise estimates of how the runtime of this OTKE compares to dot product attention? Also, how expensive is the Nystrom procedure?
>
> The runtime of OTKE on supervised protein fold classification is about 4h with 5 references and 10 supports for 100 epochs, while runtime of the dot product attention is about 3h for 100 epochs as well, on the same gpu. More estimates of the runtime for other baseline methods can be found in Table 13.
>
> On the Nyström procedure:
> - In the unsupervised setting, we used a K-means algorithm on subsampled features from the entire dataset to learn the anchor points. K-means is GPU-friendly and very fast in practice. The computation of
> the $k \times k$ projection matrix is $O(k^3)$, only computed once at train time.
> - In the supervised setting, the anchor points w are initialized with the K-means algorithm and then optimized with back-propagation as other parameters. The embedding given by Nyström is similar to a fully connected layer with the computation of the projection matrix for every minibatch, which is not a bottleneck as $k$ is small in this case. More discussion on Nyström procedure can be found in [Chen et al 2019a,b].
>
> > Misc: This is a bit of nitpicking, but the authors are a bit lazy while citing, and in several places just cite the textbooks for OT and Kernels (which is not a problem). However, it is useful for the reader to additionally have the accurate references as well, e.g. for Sinkhorn’s algorithm perhaps also cite Sinkhorn & Knopp (1967) and Cuturi (2013); for OT theory, Villani 2008 etc.
>
> We agree with this point and will make the references more accurate.

---

### Official Review · AnonReviewer4 · 2020-11-01
**A well-motivated novel embedding/pooling architecture with somewhat weak results**

**Rating:** 6
**Confidence:** 3

**Review:**

This paper proposes a kernel embedding for a set/sequence of features, based on the optimal transport distance to a set of references, leading to a fixed-dimensional embedding of variable length sequences.
The set of references can be obtained as cluster centers over the full dataset (unsupervised), or learned based on a downstream objective (supervised).
The method is somewhat related to a single layer of self-attention with an optimal transport map instead of dot product attention.
Experiments on classifying variable-length sequences are presented between protein, chromatin and NLP sentiment classification.

I recommend weak accept because (+) a well-motivated novel embedding/pooling architecture with (-) somewhat weak, initial proof-of-concept style results and (-) somewhat strange positioning against self-supervised methods and self-attention.

Strenghts:
* Well-motivated novel layer for embedding a variable-size set or sequence to a *fixed dimensional* embedding space (a point that's not directly apparent though, consider emphasizing this point)
* Relatively elegant formulation grounded in OT and kernel methods.
* The same framework allows an unsupervised and supervised variant.

Weaknesses:
* Experimental results are not very strong. For protein fold classification, I expect a comparison to transformer-based methods on proteins (eg ESM-1, https://github.com/facebookresearch/esm), simply average-pooled over the full sequence. For sentiment analysis, it is cool to see the unsupervised/linear probe on fixed features, beating the frozen BERT - however not competitive to BERT+finetuning which is how it is usually done.
* Given weak results, the impact of this method is not very clear. It is only shown on shallow single-layer OTKE, without clear discussion how this could be extended to deeper architectures.
* Given the shallow architecture, the paper is somewhat mispositioned, given the (a) introduction calling out limitations of self-supervised transformers and (b) comparison to self-attention in Sec 4: this suggests inserting the OTKE as a drop-in replacement for the dot product in a transformer model, trained with a self-supervised objective like MLM. Given the framing it seems such a logical step, it surprises me to see at least a discussion of it missing. This would be akin to recent work [1,2] of approximating self-attention by mapping to a fixed-dimensional space.
Alternatively, it seems like the method could be applied on top of strong/SotA MLM-transformer embeddings to enable better pooling than just [CLS] token or mean pooling (both protein fold prediction and sentiment classification).

Minor comments / questions:
* The labeling "unsupervised" in Table 2 and 4 is somewhat confusing/misleading, rather I'd name it "linear probe" or "supervised finetuning". Maybe re-clarify in intro of 5.1 "supervised == learning the z also"?
* What are the input features for protein fold classification? it is just mentioned they are 45-dimensional, not what is their nature (amino acid identity + pssm?)
* On end-to-end learning by back-propping through unrolled sinkhorn (Sec 3.4): this seems to be the method introduced in [3]? I think this paper should be cited in this context.
* For clarity: In both abstract, end of introduction, start of Sec 3.1 and Sec3.3, it is simply not clear to the reader what is the input/output of your method: a phrase like this would be much needed: "an embedding+pooling layer which aggregates a variable-size set or sequence of features to a fixed-size embedding".


- [1] arXiv:2006.03555 Choromanski et al. Masked Language Modeling for Proteins via Linearly Scalable Long-Context Transformers
- [2] arXiv:2006.04768 Wang et al. Linformer: Self-Attention with Linear Complexity
- [3] Genevay, A., Peyr´e, G., Cuturi, M., et al. (2018). Learning Generative Models with Sinkhorn Divergences. In AISTATS.



----
### EDIT: reply to authors' response
I thank the authors for their in-depth response and revision. The additional results comparing to pre-trained features definitely adds perspective, and it shows OTKE giving a very modest boost indeed, mostly over the weaker input features. The revision of the paper also improves the clarity, but my comment around Sec4 remains.
I stand by my original score for the 3 reasons I had originally listed.

---

> ### Author Response · Authors · 2020-11-17
> **individual response to R4**
>
> We thank the reviewer for his/her detailed feedback.
>
> > For protein fold classification, I expect a comparison to transformer-based methods on proteins (eg ESM-1, [...]
>
> We thank the reviewer for pointing out this work. We were a bit surprised by the request to compare to ESM as the code of ESM was released less than a month before the ICLR deadline. Nevertheless, we admit that this is a very good suggestion. We have therefore added several baselines for the SCOP experiments:
> * using ESM-1, pretrained on 250 millions sequences, with mean pooling, followed by a linear classifier. As we do not have the computational ressources to fine-tune ESM-1_t34, we only train a linear layer on top of the extracted features. We tried finetuning ESM-1_t6 but did not get any improvement.
> * using the same ESM-1 model, when replacing mean pooling by our OTKE layer. To be fair, we consider only unsupervised OTKE here, and also only train the linear layer.
>
> The results are presented in Table C below. In contrast, our best OTKE model (mentioned in general comments) achieves 91.24/96.77/97.78, which yields the following conclusions
> * training huge self-supervised learning models on large datasets is effective, as the performance of ESM-1_t34 is impressive. Note however that the context is different, as ESM-1_t34 admits more than 2500 times more parameters (trained with self-supervision on large-scale external data) than our single-layer OTKE model (260k parameters vs 670M for ESM-1_t34).
> * our single-layer OTKE outperforms ESM-1_t6 (43M parameters).
> * plugging unsupervised OTKE on top on ESM-1 brings some benefits, mostly for ESM-1_t6.
>
> This confirms that competitive advantages of ESM and OTKE are different and complementary (see also general comments on positioning), and that pre-training with self-supervision a large model including OTKE on large data sets would be interesting for fair comparison.
>
> **Table C**
>
> | model	| #params | mean pooling | with unsupervised OTKE |
> | -- | -- | -- | -- |
> | esm1_t6_43M_UR50S | 43M | 84.01/93.17/95.07 | 85.91/93.72/95.30 |
> | esm1_t34_670M_UR50S | 670M | 94.95/97.32/97.91 |95.22/97.32/98.03 |
>
>
> > For sentiment analysis, it is cool to see the unsupervised/linear probe on fixed features, beating the frozen BERT - however not competitive to BERT+finetuning which is how it is usually done.
>
> For NLP, and given its positioning (see general comments), we have chosen to compare our work to other models for sets, which we outperform in this setting. We believe our method to be of interest in a constrained ressource context, where it is not possible to fine-tune a whole transformer architecture. We indeed do not claim that the method will be interested for all NLP tasks (we rather indeed claim that the results on NLP should be seen as a proof of concept). For instance, many NLP tasks (e.g, tasks of GLUE) have a different setting from the one for which our method was initially built: sequences are shorter and/or there are more data. We choose SST-2 because it is a sentence classification task with relatively long sequences with a moderate amount of data.
>
> > It is only shown on shallow single-layer OTKE, without clear discussion how this could be extended to deeper architectures.
>
> see general comments.
>
> >  Given the shallow architecture, the paper is somewhat mispositioned, [...] , it seems like the method could be applied on top of strong/SotA MLM-transformer embeddings ...
>
> We agree with many of these remarks, see general comments.
>
> > The labeling "unsupervised" in Table 2 and 4 is somewhat confusing/misleading, ...
>
> Here, "unsupervised" refers to "representation learning without supervision" (z, but also Nyström anchors w). In this setting, only the classifier is learned with supervision. The same terminology is used in previous works on which we rely [Mairal 2016, Chen et al. 2019 a;b]. We propose to keep those terms and clarify their meaning in the paper.
>
> > What are the input features for protein fold classification? it is just mentioned they are 45-dimensional, not what is their nature (amino acid identity + pssm?)
>
> The input features consist of a 20-dimensional one-hot encoding of the sequence, a 20-dimensional position-specific scoring matrix (PSSM) representing the profile of amino acids computed by PSI-BLAST against the nr90 database, a 3-class secondary structure represented by a one-hot vector and a 2-class solvent accessibility. More details are provided in Appendix C.3.
>
> > On end-to-end learning by back-propping through unrolled sinkhorn (Sec 3.4): this seems to be the method introduced in [3]?
>
> [3] also uses such an approach. We thank the reviewer for this comment and will add the reference.
>
> > For clarity: In both abstract, end of introduction, start of Sec 3.1 and Sec 3.3, it is simply not clear to the reader what is the input/output of your method: a phrase like this ....
>
> We agree with the reviewer and will add a sentence to clarify this.

---

### Author Response · Authors · 2020-11-17
**Rebuttal: general comments**

First, we would like to thank all reviewers for their detailed feedback. We answer general comments here and provide individual answers to each reviewer. We are working on a revision of the paper taking into account these comments, which we will upload as soon as possible.

**Positioning with respect to self-supervised learning and self-attention (R4, R1)**

We agree with R4 that our introduction is misleading regarding the positioning of our paper. Our approach should be simply seen as a _feature aggregation method_, with few parameters, providing a fixed-sized embedding for sets of possibly varying sizes. We illustrate this method on sequence data, even though other modalities may be used in principle (e.g., point clouds, aggregation of local features in graphs). Therefore,
- _on self-supervision_:  our approach should indeed not be positioned against self-supervision models. In fact, it may be plugged in sota self-supervised models trained on large unannotated corpus, when it is possible, as suggested by R4 (see additional experiment in response to R4).
- _on self-attention_: as shown in Sec. 4, there is a relation between our work and self attention, but our goal (providing fixed-size embeddings for sets) is different from approximating classical self-attention matrices. Nevertheless, we agree that references [1,2] from R4 and mentioned by R1 are relevant, as they use low-rank approximations of a (kernel) attention matrix, whereas our embedding provides a factorization of the approximate transport plan $P_z$.

**On multi-layer OTKE (R3,R4)**

This is indeed a very natural question, which can addressed from two points of views:
* plugging OTKE as the last aggregation layer of a deep architecture is feasible and seems to work in practice (see experiment in response to R4). See also the DeepSea experiment, where we replace the last convolutional layers of the DeepSea CNN by OTKE.
* building a multi-layer OTKE is an interesting research direction, but it is not straightforward and our early attempts have been inconclusive. One possible reason is that our method performs well when aggregating features from large sets of varying sizes (which is well adapted to the last layer of a deep architecture) and it is not clear how to find a right definition of intermediate feature aggregation in a multi-layer OTKE model. Note that for DeepSEA, our model with single-layer OTKE already outperforms a multi-layer CNN.

**On architecture choices for OTKE (R1,R3)**

We agree that a summary of the impact of hyper-parameter choices is currently missing in the paper. In short,
- _Number of references $q$_: for biological sequences, a single reference was found to be enough in the unsupervised case (see Table 11). In the supervised setting, Table 13 suggests that using $q=5$ provides slightly better results but $q=1$ remains a good baseline. Table A (added below for this rebuttal) is more exhaustive than Table 13 and suggests that the sensitivity to number of references is moderate.
- _Number of supports p in a reference_: Table 11 and Table A suggest that the sensitivity of the model to the number of supports is also moderate, but we agree that finding automatically the optimal number of supports in a principled way is an important open problem.
- _Nyström anchors_: an anchor point can be seen as a neuron in a feed-forward neural network (see last line of page 4). For unsupervised settings, the more anchors, the better the approximation of the kernel matrix, and the performance saturates when this number is large enough, as illustrated below in Table B, added for this rebuttal. In the supervised setting, the optimal number of anchors points is much smaller than unsupervised, as also observed in [Chen et al. 2019a], Figure 6.
- _bandwidth $\sigma$ in gaussian kernel_:  $\sigma$ for the features was chosen as in [Chen et al. 2019a] and we did not try to optimize it for the submission, as it seemed to already provide good results. Nevertheless, slightly better results can be obtained when tuning this parameter. For instance, $\sigma=0.6$ yields better validation and test accuracy on SCOP (91.24$\pm$0.28/96.77$\pm$0.17/97.78$\pm$0.19 vs  88.7/95.9/97.3 in Table 2)

**Table A (same as Table 11 on SCOP, but for supervised OTKE)**

| $q \times p$ | 10 | 50 |100| 200 |
|-------------------| ----| ----|-----|------|
| 1  | 88.3/95.5/97.0 | 88.4/95.8/97.2 | 87.1/94.9/96.7 | 87.7/94.9/96.3 |
| 2  | 87.8/95.8/97.0 | 89.6/96.2/97.5 | 86.5/94.9/96.6 | 87.6/94.9/96.3 |
| 5  | 87.0/95.1/96.7 | 88.8/96.0/97.2 | 87.4/95.4/97.0 | 87.4/94.7/96.2 |
| 10| 84.5/93.6/95.6 | 89.8/96.0/97.2 | 88.0/95.7/97.0	| 85.6/94.4/96.1 |

**Table B (additional results for unsupervised OTKE on SCOP when varying the number of anchor points)**

| Method | Accuracies |
| ----------- | ------ |
| OTKE-1024	| 85.8/95.3/96.8 |
| OTKE-2048	| 86.6/95.9/97.2 |
| OTKE-3072	| 87.8/96.1/97.4 |

---

### Author Response · Authors · 2020-11-20
**revision is uploaded**

We have uploaded a revision of the paper, which takes into account the comments below. We would be happy to make further modifications, if the reviewers want to provide additional feedback.

---

### Comment · ~Soheil_Kolouri1 · 2021-02-16
**Relationship to the Linear Optimal Transport**

Congratulations on the acceptance of the paper, and thank you for the excellent work. The concept of pooling with OT using a reference measure is interesting. We have also used this concept in our ICLR2021 paper, titled "Wasserstein Embedding for Graph Learning," for pooling with the application of learning from graphs: https://openreview.net/forum?id=AAes_3W-2z .

One important point (maybe for the sake of completeness) is that the $W_2^z$ distance in your Eq. (4) is equivalent to the Linear Optimal Transport (LOT) distance introduced in:

Wang, W., Slepčev, D., Basu, S., Ozolek, J.A. and Rohde, G.K., 2013. A linear optimal transportation framework for quantifying and visualizing variations in sets of images. International journal of computer vision, 101(2), pp.254-269.

and more recently closely studied in:

Moosmüller, C. and Cloninger, A., 2020. Linear Optimal Transport Embedding: Provable fast Wasserstein distance computation and classification for nonlinear problems. arXiv preprint arXiv:2008.09165.

I thought these relationships could be interesting to the authors and the readers.

---

> ### Comment · ~Grégoire_Mialon1 · 2021-02-19
> **Thank you!**
>
> Hi Soheil,
>
> Thank you for your comments and references, which we will add in our bibliography. We look forward to discuss our respective works at the conference!

---

### Decision · Program_Chairs · 2021-01-07
**Final Decision**

**Decision:**

Accept (Poster)

**Comment:**

All reviewers agreed that the paper proposes some interesting and novel ideas on the use of OT for pooling. It also provides some nice insights and strong experimental results. As suggested by one of the reviewer, a discussion about the impact of the number of references may be of interest though.